# Mechanisms of PP2A-Ankle2 dependent nuclear reassembly after mitosis

Jingjing Li[1,2], Xinyue Wang[1], Laia Jordana[1,2], Éric Bonneil[1], Victoria Ginestet[1], Momina Ahmed[1], Mohammed Bourouh[1], Cristina Mirela Pascariu[1], T Martin Schmeing[3], Pierre Thibault[1,4], Vincent Archambault[1,2]*

[1]Institute for Research in Immunology and Cancer, Université de Montréal, Montreal, Canada; [2]Département de biochimie et médecine moléculaire, Université de Montréal, Montreal, Canada; [3]Department of Biochemistry, McGill University, Montreal, Canada; [4]Département de chimie, Université de Montréal, Montreal, Canada

*For correspondence:
vincent.archambault.1@
umontreal.ca

## eLife Assessment

This is an **important** study that reports the mechanism by which Ankle2 (LEM4 in humans) interacts with and recruits PP2A and the ER protein Vap33 to promote BAF dephosphorylation and mediate nuclear membrane reformation, using *Drosophila* as their model. Using Ankle2 mutants, they find that the ER protein Vap33 is key for the normal interphase localisation of Ankle2/LEM4 and also impacts on the function of Ankle2/LEM4 during mitosis. The conclusions on the subcellular localization of Ankle2 are drawn from overexpression of constructs. Overall, the authors use a variety of complementary techniques and provide **convincing** evidence to support the claims and advance our knowledge in the field of mitosis and nuclear envelope biology.

**Abstract** In animals, mitosis involves the breakdown of the nucleus. The reassembly of a nucleus after mitosis requires the reformation of the nuclear envelope around a single mass of chromosomes. This process requires Ankle2 (also known as LEM4 in humans) which interacts with PP2A and promotes the function of the Barrier-to-Autointegration Factor (BAF). Upon dephosphorylation, BAF dimers cross-bridge chromosomes and bind lamins and transmembrane proteins of the reassembling nuclear envelope. How Ankle2 functions in mitosis is incompletely understood. Using a combination of approaches in *Drosophila*, along with structural modeling, we provide several lines of evidence that suggest that Ankle2 is a regulatory subunit of PP2A, explaining how it promotes BAF dephosphorylation. In addition, we discovered that Ankle2 interacts with the endoplasmic reticulum protein Vap33, which is required for Ankle2 localization at the reassembling nuclear envelope during telophase. We identified the interaction sites of PP2A and Vap33 on Ankle2. Through genetic rescue experiments, we show that the Ankle2/PP2A interaction is essential for the function of Ankle2 in nuclear reassembly and that the Ankle2/Vap33 interaction also promotes this process. Our study sheds light on the molecular mechanisms of post-mitotic nuclear reassembly and suggests that the endoplasmic reticulum is not merely a source of membranes in the process, but also provides localized enzymatic activity.

## Introduction

In animal cells, the disassembly of the nucleus during mitosis allows the segregation of the genetic material via the spindle apparatus, composed of cytoplasmic microtubules and centrosomes. However, this process of open mitosis complicates the transition from mitosis to interphase, as a new nucleus

must reassemble around each set of segregated chromosomes. The mechanisms of nuclear reassembly after mitosis remain incompletely understood (*Hampoelz and Baumbach, 2023*; *Kono and Shimi, 2024*; *Li et al., 2024*; *Schellhaus et al., 2016*; *Ungricht and Kutay, 2017*).

The nuclear envelope (NE) is a double membrane that is topologically continuous with the endoplasmic reticulum (ER) (*Deolal et al., 2024*; *Ungricht and Kutay, 2017*). The NE is associated with multiple structural proteins that connect it to chromatin and cytoskeleton. Lamins form a semi-rigid cage-like network between chromatin and the inner nuclear membrane while the Barrier-to-Autointegration Factor (BAF) interacts with DNA, Lamins, and transmembrane LEM-Domain proteins within the inner nuclear membrane (*Sears and Roux, 2020*). Nuclear pore complexes composed of multiple nucleoporins ensure transport across the NE and also interact with structural cytoplasmic and nucleoplasmic proteins (*Goldberg, 2017*; *Petrovic et al., 2022*; *Shevelyov, 2023*). In addition, the LINC complex spans the entire NE to connect chromatin to the cytoskeleton and relay mechanotransduction (*Hieda, 2019*). During mitotic entry, the phosphorylation of many of these proteins disrupts their interactions, resulting in nuclear envelope breakdown (NEB) (*Archambault et al., 2022*). BAF is phosphorylated by Vaccinia-Related Kinases (VRKs, NHK-1/Ballchen in *Drosophila*) at its N-terminus, which impedes its binding to DNA (*Lancaster et al., 2007*; *Nichols et al., 2006*). Similarly, CDK1, PKC, and PLK1 phosphorylate Lamins A/C and B, disrupting their polymerization (*Liu and Ikegami, 2020*; *Velez-Aguilera et al., 2020*). As a result of NEB, NE membranes retract into the ER during mitosis.

Correct reassembly of the nucleus after mitosis is critical for cell viability and physiology. During this process, segregated chromosomes gather together and ER membranes are recruited around chromosomes to rebuild the NE (*Deolal et al., 2024*; *Schellhaus et al., 2016*; *Ungricht and Kutay, 2017*). Failure to group chromosomes into a single nucleus results in micronuclei that are prone to rupture and DNA damage (*Guo et al., 2020*). The timely reformation of the nuclear envelope is essential for reestablishing nucleocytoplasmic transport that is needed for the entire gene and protein expression process, including transcription, mRNA export, and ribosome biogenesis (*Alberts et al., 2015*). Protein Phosphatases 1 (PP1) and 2 A (PP2A) dephosphorylate several proteins to promote their interactions and reassociation with the nuclear envelope (*Archambault et al., 2022*; *Huguet et al., 2019*). How the activities of the enzymes are coordinated in time and space is incompletely understood.

BAF plays a central role in nuclear reassembly. It forms a dimer that binds DNA after anaphase, thereby cross-linking chromosomes into a single cohesive mass (*Samwer et al., 2017*). BAF also promotes the recruitment of ER membranes through its interactions with LEM-Domain proteins, and aids in the reassembly of the Lamina (*Haraguchi et al., 2008*; *Haraguchi et al., 2001*). The dephosphorylation of BAF is required for its recruitment to segregated chromosomes (*Archambault et al., 2022*; *Sears and Roux, 2020*). The B55 regulatory subunit of PP2A promotes this regulation in various animals, including *C. elegans*, *H. sapiens,* and *D. melanogaster* (*Asencio et al., 2012*; *Mehsen et al., 2018*). In addition, the PP2A-interacting, ER-localized protein Ankle2 (also known as LEM4 and LEM-4L in humans and *C. elegans*, respectively) is also required for the recruitment of BAF to segregated chromosomes in the same three organisms (*Asencio et al., 2012*; *Li et al., 2024*; *Snyers et al., 2018*). In humans, Ankle2/LEM4 contains Ankyrin repeats, a LEM domain, a transmembrane domain, and other predicted structured and unstructured regions (*Fishburn et al., 2024*). The molecular mechanisms by which Ankle2 functions in nuclear reassembly are unclear (*Figure 1A*). *ANKLE2* function is crucial for the development of the central nervous system as mutations in this gene are associated with microcephaly (*Thomas et al., 2022*; *Yamamoto et al., 2014*). Targeting of Ankle2 by the Zika virus also causes microcephaly (*Shah et al., 2018*). In *Drosophila*, Ankle2 is required for asymmetric divisions in larval neuroblasts (*Link et al., 2019*). While Ankle2 impacts the asymmetric division protein machinery, it remains unclear whether the loss of its function in nuclear reassembly also contributes to microcephaly.

To investigate Ankle2 molecular functions, we conducted studies in *Drosophila*. We found that Ankle2 forms a complex with the PP2A structural and catalytic subunits, in a mutually exclusive manner with other known regulatory subunits. Our phosphoproteomic analysis validates that BAF dephosphorylation depends on Ankle2. We also identified and characterized a novel interaction of Ankle2 with Vap33, a VAP family protein that promotes Ankle2 localization to the ER during both nuclear reassembly and interphase. We show that the interactions of Ankle2 with PP2A and Vap33 promote BAF recruitment to chromosomes after anaphase. Moreover, the Ankle2/PP2A interaction

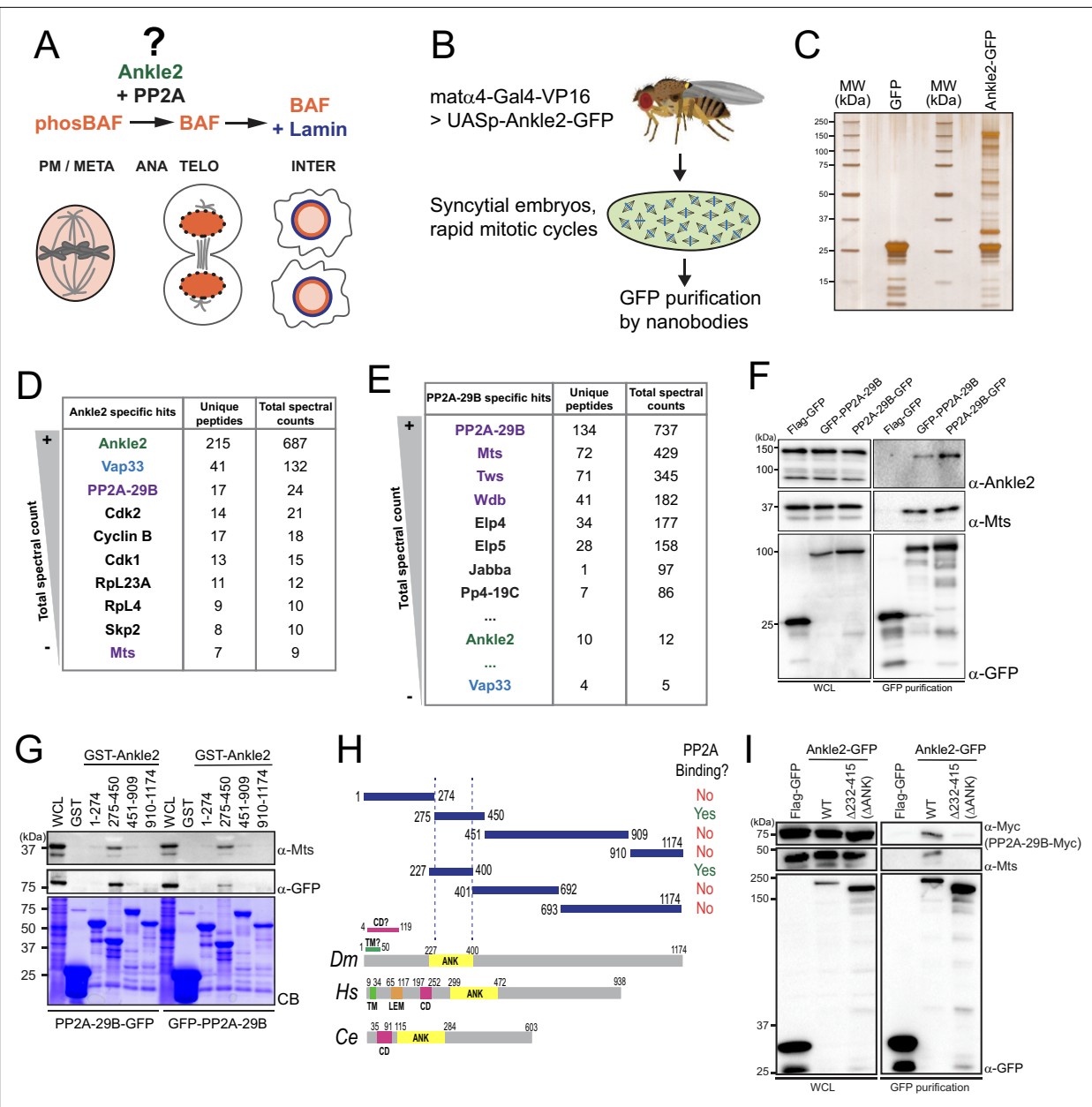

**Figure 1.** Ankle2 interacts with Protein Phosphatases 2A (PP2A) through its Ankyrin repeats region. (**A**) Ankle2 functions with PP2A to allow the dephosphorylation of Barrier-to-Autointegration Factor (BAF) and its recruitment to segregated chromosomes. BAF promotes the formation of a single nucleus by cross-bridging chromosomes, and the recruitment of Lamin and endoplasmic reticulum (ER) membranes containing LEM-Domain proteins (not shown). The molecular mechanisms involving Ankle2 in this process are incompletely understood. (**B**) Strategy for the identification of Ankle2 interactor proteins in vivo. See text for details. (**C–E**) Proteins obtained after purification of Ankle2-GFP, PP2A-29B-GFP, or GFP from embryos. (**C**) Silver-stained gel showing a fraction of the purification products. (**D**) Proteins specifically identified with Ankle2-GFP. Proteins with the highest total spectral counts are shown. (**E**) Proteins were specifically identified with PP2A-29B-GFP after purification from embryos. Proteins with the highest total spectral counts are shown. Ankle2 and Vap33 were also identified as specific interactors further down the list. Purple names: known PP2A subunits. (**F**) Ankle2 is specifically co-purified with PP2A-29B. Cells were transfected with the indicated proteins and used in GFP affinity purifications. Products were analyzed by western blots. WCL: Whole cell extracts. (**G**) A region of Ankle2 between amino-acid residues 275–450 is sufficient for interaction with PP2A. GST-fused fragments of Ankle2 produced in bacteria were used in GST-pulldowns with extracts from D-Mel cells expressing PP2A-29B-GFP or GFP-PP2A-29B. Pulled-down proteins (PP2A-29B-GFP or GFP-PP2A-29B and Mts) were detected by western blots. CB: Coomassie Blue. (**H**) Top: Summary of results from GST pulldowns testing interactions of Ankle2 fragments with PP2A. Bottom: Primary structures of Ankle2 from *D. melanogaster* (*Dm*), *H. sapiens* (*Hs*), and *C. elegans* (*Ce*). showing known motifs or domains. ANK: Ankyrin domain region; TM: Trans-Membrane motif: LEM: LEM domain. CD: Caulimovirus Domain. (**I**) The Ankyrin domain region of Ankle2 between amino-acid residues 232–415 is required for interaction with PP2A. Cells

*Figure 1 continued on next page*

*Figure 1 continued*

were transfected with the indicated proteins and used in GFP affinity purifications. Products were analyzed by western blots. WCL: Whole cell extracts. Numerical data are available in *Figure 1—source data 1*, *Figure 1—source data 2*.

The online version of this article includes the following source data and figure supplement(s) for figure 1:

**Source data 1.** Proteins identified from affinity purifications of Ankle2-GFP from embryos.

**Source data 2.** Proteins identified from affinity purifications of PP2A-29B-GFP from embryos.

**Source data 3.** Figures with uncropped western blots and gels annotated.

**Source data 4.** Original files for western blots and gels.

**Figure supplement 1.** Ankle2 interacts with Protein Phosphatases 2A (PP2A) through its Ankyrin repeats region.

**Figure supplement 1—source data 1.** Proteins identified from affinity purifications of GFP-fused Ankle2 from D-Mel cells.

**Figure supplement 1—source data 2.** Proteins identified from affinity purifications of GFP-fused PP2A-29B from D-Mel cells.

**Figure supplement 1—source data 3.** Figures with uncropped western blots and gels annotated.

**Figure supplement 1—source data 4.** Original files for western blots and gels.

is essential for post-mitotic nuclear reassembly and development in vivo. We propose that Ankle2 functions as a novel PP2A regulatory subunit required for BAF dephosphorylation, and that Vap33-dependent anchoring of Ankle2 to the ER promotes this function in a localized manner during post-mitotic nuclear reassembly.

## Results

### Ankle2 interacts with PP2A through its Ankyrin domain

To investigate the molecular functions of Ankle2, we sought to identify its interaction partners. First, we generated D-Mel (D.mel2) stable cell lines expressing GFP-fused Ankle2 (N or C terminal). We performed GFP affinity purifications on the lysate of these cells and identified co-purified proteins by mass spectrometry. Cells expressing Flag-GFP were used as controls. We found that PP2A-29B (PP2A-A, structural) and Microtubule Star (Mts, PP2A-C, catalytic) were specifically co-purified with both GFP-Ankle2 and Ankle2-GFP (*Figure 1—figure supplement 1A, B*, *Figure 1—figure supplement 1—source data 1*). To test if this association occurs in vivo, we generated transgenic flies for the expression and purification of Ankle2-GFP in early embryos (*Figure 1B*). The maternal driver matα4-GAL-VP16 was used to activate the expression of UASp-Ankle2-GFP in late oocytes and syncytial embryos. Embryos expressing GFP alone were used as controls. Again, we found PP2A-29B and Mts specifically co-purified with Ankle2-GFP (*Figure 1C, D*, *Figure 1—source data 1*). Reciprocally, endogenous Ankle2 was co-purified with PP2A-29B-GFP from cells in culture and embryos, along with the regulatory subunits Tws, Wdb, and Wrd (*Figure 1E, F*, *Figure 1—figure supplement 1C, D*, *Figure 1—source data 2*, *Figure 1—figure supplement 1—source data 2*). Thus, the interaction of Ankle2 with PP2A is conserved between *Drosophila* and humans (*Asencio et al., 2012*).

To map the region of Ankle2 that binds PP2A, we used a GST pulldown assay. We produced GST-fused fragments of Ankle2 covering its entire sequence in *E. coli* and tested their ability to pulldown PP2A-29B (with GFP-fused to either end) and Mts from D-Mel cells. We initially found that a fragment comprising amino acid (a.a.) residues 275–450 was sufficient to strongly associate with PP2A-29B and Mts, while fragments 1–274, 451–909, or 910–1174 showed no or very weak association with these proteins (*Figure 1G, H*). The 275–450 region comprises most of the Ankyrin domain. This domain was recently proposed to span a.a. residues 229–399 (*Fishburn et al., 2024*), and our inspection of a structural model generated by AlphaFold3 (*Abramson et al., 2024*) suggested a.a. residues 227–400. We found that the Ankyrin domain (a.a. 227–400 tested) was sufficient for association with PP2A (*Figure 1—figure supplement 1E*, *Figure 1H*). To test if the Ankyrin domain region is required for the interaction of Ankle2 with PP2A, we used a co-purification assay in D-Mel cells. Ankle2-GFP WT or with a deletion of the Ankyrin domain were purified and the presence of co-purified PP2A-29B-Myc and Mts was probed by western blot. In this assay, we deleted a region comprising the predicted Ankyrin domain plus a few additional downstream residues showing conservation with human Ankle2 and *C. elegans* Lem-4L; (ΔANK: residues 232–415 deleted; *Figure 1—figure supplement 1F*). We found that PP2A-29B-Myc and Mts were co-purified with Ankle2$^{WT}$-GFP but not with Ankle2$^{\Delta ANK}$-GFP

(*Figure 1I*). We conclude that the region of Ankle2 comprising the Ankyrin domain is necessary and sufficient to bind the PP2A-29B/Mts complex in *Drosophila*.

## Ankle2 functions as a PP2A regulatory subunit to promote BAF dephosphorylation

We hypothesized that Ankle2 functions as a regulatory protein required for PP2A to dephosphorylate some of its substrates. The knockdown of LEM-4L or Ankle2 results in the hyperphosphorylation of BAF in its N-terminus in *C. elegans* or *Drosophila* cells, respectively (*Asencio et al., 2012*; *Li et al., 2024*). To identify sites that become hyperphosphorylated in the absence of Ankle2 at a proteome-wide level, we used a phosphoproteomic approach (*Emond-Fraser et al., 2023*). We transfected D-Mel cells with dsRNA against Ankle2 or dsRNA Non-Target (NT). Four days later, cells were lysed, proteins were digested with trypsin, and phosphopeptides were enriched, and analyzed by quantitative mass spectrometry. Among hyperphosphorylated peptides upon Ankle2 depletion, we found the expected BAF peptide phosphorylated at Thr4 and Ser5 (*Figure 2A*). Phosphorylation at the orthologous sites in human BAF was shown to negatively regulate its ability to interact with DNA, and possibly with LEM-Domain proteins (*Nichols et al., 2006*). Our observation of these Ankle2-dependent sites in BAF validates our approach since BAF is the only reported protein whose phosphorylation state depends on Ankle2 (*Asencio et al., 2012*; *Li et al., 2024*). Moreover, BAF could be the only obligatory substrate of Ankle2-dependent dephosphorylation for cell proliferation as lowering the dose of the BAF kinase NHK-1/Ballchen or expression of an unphosphorylatable mutant form of BAF rescues wing development defects caused by the partial depletion of Ankle2 (*Li et al., 2024*). Nevertheless, we identified several additional hyperphosphorylated sites in other proteins upon Ankle2 knockdown (*Figure 2A*, *Figure 2—source data 1*). Although it is likely that some of the hyper- and hypophosphorylated sites observed reflect indirect alterations of cell physiology upon Ankle2 depletion, our results point at several candidate proteins whose dephosphorylation may depend on Ankle2.

We hypothesized that Ankle2 functions as a regulatory subunit of PP2A. Interestingly, the structural (PP2A-29B) and catalytic (Mts) subunits of PP2A were detected in complex with Ankle2, while none of the known regulatory subunits of PP2A were detected (Tws, Wdb, Wrd, or PR72) (*Figure 1D*, *Figure 1—figure supplement 1B*, *Figure 1—source data 1*, *Figure 1—figure supplement 1—source data 1*). To test if Ankle2 interacts with PP2A as a regulatory subunit, we used a competition assay (*Figure 2B*). We co-transfected cells with Ankle2-Myc and PP2A-29B-GFP or FLAG-GFP. We then proceeded to GFP affinity purifications. As expected, Ankle2-Myc was co-purified specifically with PP2A-29B-GFP. The PP2A regulatory subunit Tws was also co-purified specifically with PP2A-29B-GFP. To test if Tws and Ankle2 are mutually exclusive in the PP2A complex, we co-transfected increasing amounts of Myc-Tws plasmid along with Ankle2-Myc and PP2A-29B-GFP. As a result, increasing amounts of Myc-Tws were co-purified with PP2A-29B-GFP. Conversely, decreasing amounts of Ankle2-Myc were co-purified with PP2A-29B-GFP as the amount of Myc-Tws increased. These results suggest that Tws competes with Ankle2 for binding within the PP2A holoenzyme and reinforce the idea that Ankle2 functions as a *bona fide* regulatory subunit of PP2A.

To contextualize and interpret our results, we generated structural predictions of relevant co-complexes using AlphaFold3 (*Abramson et al., 2024*). In models of a complex of *Drosophila* Ankle2, PP2A-29B, and Mts, the Ankyrin domain consistently directly interacts with Mts (*Figure 2C* shows a representative model), consistent with our pulldown results (*Figure 1G*, *Figure 1—figure supplement 1E*). Interestingly, these models also predict that additional regions of Ankle2 engage in interactions with PP2A. A region previously referred to as the Caulimovirus Domain (CD, *Figure 1H*) and proposed to mediate an interaction between human Ankle2 and PP2A (*Fishburn et al., 2024*) is indeed predicted to interact with Mts in our models, on the other side from where the Ankyrin domain binds. Moreover, the modeling shows an adjacent uncharacterized domain (UNC) that is in direct contact with PP2A-29B. However, no interaction with Mts or PP2A-29B was experimentally detected for our GST-fused fragments comprising these regions alone in pulldown assays. It is possible that the UNC and CD domains interact with the PP2A subunits with low affinity. It is also possible that the fragment tested requires additional components to fold into a form that is competent for PP2A binding; in particular, AlphaFold3 predictions show that the complete UNC domain may be made of several segments distal in primary sequence (a.a. residues ~401–467 + 601–682 + 987–1050). In any case, the deletion of the Ankyrin domain of Ankle2 strongly abrogated its formation of a complex

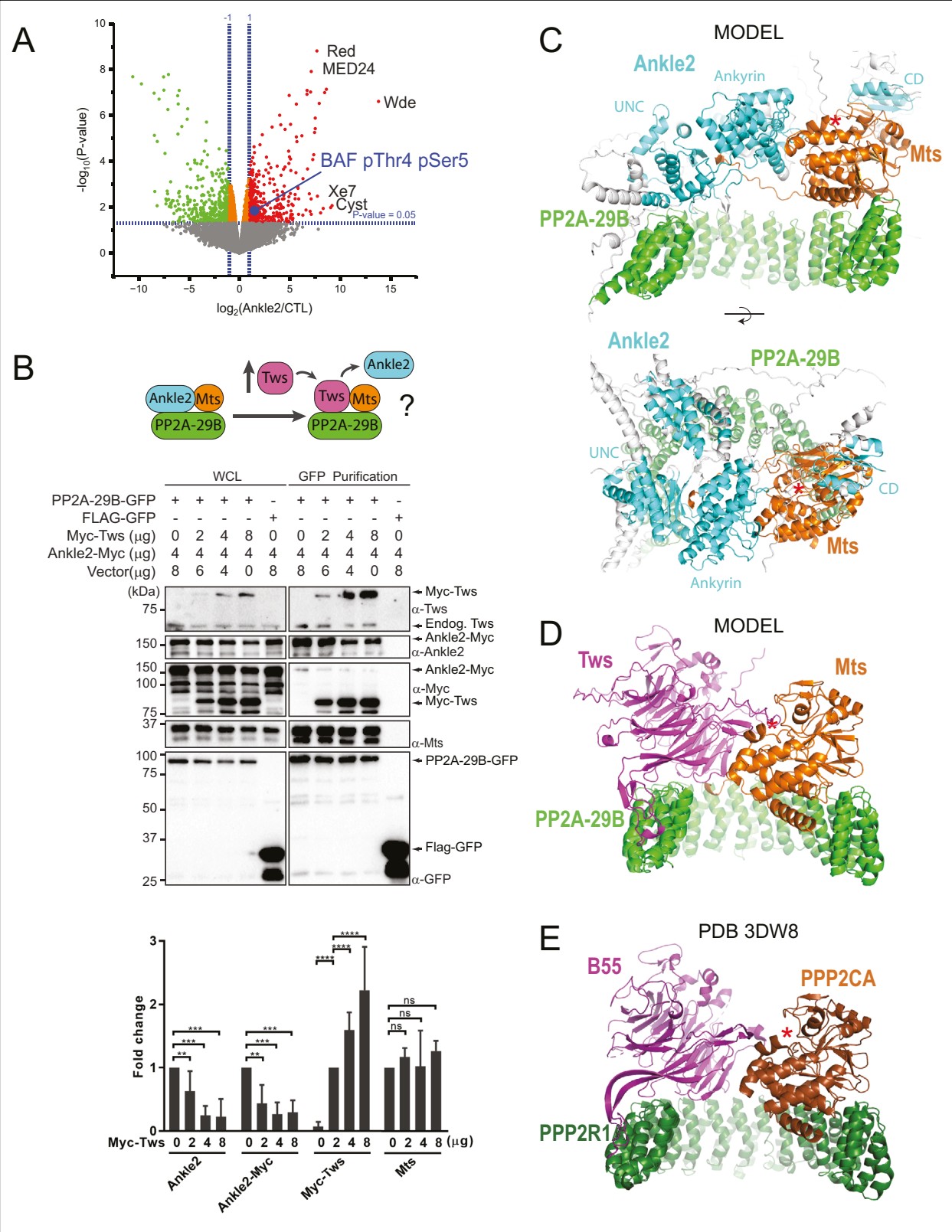

**Figure 2.** Protein Phosphatases 2A (PP2A)-Ankle2 promotes Barrier-to-Autointegration Factor (BAF) dephosphorylation. (**A**) Phosphoproteomic analysis identified BAF as being hyperphosphorylated at Thr4 and Ser5 upon Ankle2 depletion. D-Mel cells were transfected with dsRNA against Ankle2 or non-target dsRNA (against bacterial KAN gene). Phosphopeptides from tryptic digests were analyzed quantitatively by mass spectrometry. A few additional significantly hyperphosphorylated proteins are labeled as examples (see *Figure 2—source data 1* for the full list). Red, green, and orange

*Figure 2 continued on next page*

*Figure 2 continued*

dots represent peptides with significant (p<0.05) hyperphosphorylation, hypophosphorylation, and little change, respectively. Gray dots indicate peptides with changes below statistical significance. (**B**) Competition assay. Top: schematic hypothesis. If Ankle2 occupies the position of a PP2A regulatory subunit, its presence in the PP2A holoenzyme may be outcompeted by Tws (PP2A regulatory subunit). Bottom: the interaction between PP2A-29B-GFP and Ankle2-Myc was monitored by a GFP affinity co-purification assay. Increasing amounts of Myc-Tws plasmid were co-transfected. As a result, increasing amounts of Myc-Tws, and decreasing amounts of Ankle2-Myc, are co-purified with PP2A-29B-GFP. The amounts of co-purified Mts are unchanged. Averages of three experiments are shown. All error bars: S.D. **p<0.01, ***p<0.001 **** p<0.0001, ns: non-significant from paired t-tests (**C**) AlphaFold3 predicted a model of a complex between *Drosophila* Ankle2, PP2A-29B, and Mts. Residues of Ankle2 with confidence scores below 0.8 are coloured gray. (**D**) Predicted model of a complex between *Drosophila* Tws, PP2A-29B and Mts. (**E**) Crystal structure of a complex between human PPP2R1A, PPP2CA, and B55 (PDB 3DW8) (*Xu et al., 2008*). Red asterisks denote the positions of the phosphatase catalytic site. Numerical data are available in *Figure 2—source data 1*, *Figure 2—source data 2*.

The online version of this article includes the following source data for figure 2:

**Source data 1.** Phosphopeptide identification and quantification from cells after Ankle2 or control RNAi.

**Source data 2.** Numerical data is used to make graphs.

**Source data 3.** Figures with uncropped western blots annotated.

**Source data 4.** Original files for western blots.

with Mts and PP2A-29B, indicating that it plays a predominant role in the formation of the complex. Importantly, these models display Ankle2 occupying an overlapping position to that of Tws in an analogous PP2A-Tws model which itself consistent with a crystal structure of human PP2A-B55 (*Xu et al., 2008*; *Figure 2C–E*). These models account for the mutually exclusive binding of Ankle2 or Tws to the PP2A-29B/Mts complex, and are consistent with the function of Ankle2 as a regulatory subunit of PP2A.

## Ankle2 interacts with Vap33 and they colocalize at the endoplasmic reticulum

Besides PP2A, we discovered that Ankle2 interacts with Vap33, an integral protein of the endoplasmic reticulum (ER) and a member of the VAP family (*Figure 1D*). Mass spectrometry data suggested that Vap33 was co-purified at the highest levels among Ankle2 interactors. Ankle2 was previously reported to localize at the ER and the NE in *Drosophila* (*Link et al., 2019*). We found that Ankle2-RFP co-localizes with GFP-Vap33 in D-Mel cells (*Figure 3A* and *Video 1*). This co-localization is clearest during mitosis, where both proteins are enriched around the presumed spindle in metaphase and around segregated chromosomes in late anaphase/telophase, when NE reassembly takes place. To examine the localization of Ankle2 relative to Vap33 in vivo, we generated transgenic flies for the inducible expression of Ankle2-GFP and RFP-Vap33. We expressed the fusion proteins in syncytial embryos and studied their localization. We found that both proteins co-localized throughout the cell cycle (*Figure 3B* and *Video 2*). In interphase, they localized to the NE and ER membranes. During mitosis, Ankle2-GFP and RFP-Vap33 became enriched on the spindle envelope until they wrapped around newly forming nuclei in telophase. In addition, both proteins were also enriched at the midbody of the central spindle, consistent with the recruitment of ER membranes at this site (*Bobinnec et al., 2003*). We were unable to examine the localization of endogenous Ankle2 because the antibodies that we generated failed to reliably detect Ankle2 in immunofluorescence. For the remainder of our study, we used overexpressed Ankle2-GFP, which may not perfectly reflect the localization and function of endogenous Ankle2. However, Ankle2-GFP is functional as it can rescue the phenotypes observed when endogenous Ankle2 is depleted (see below).

## Ankle2 interacts with Vap33 through FFAT motifs

We confirmed the Ankle2/Vap33 interaction in a reciprocal co-purification assay. We found that endogenous Ankle2 was specifically co-purified with GFP-Vap33 or Vap33-GFP from D-Mel cells (*Figure 4A*). To identify the region of Ankle2 responsible for its interaction with Vap33, we used the GST pulldown described above, probing for the interaction of Vap33-Myc with GST-fused fragments of Ankle2. We found that the C-terminal region of Ankle2 (residues 910–1174) is sufficient to interact with Vap33 (*Figure 4B*). The VAP family of proteins is anchored to the ER via a transmembrane domain and exposes a globular MSP domain to the cytoplasm (*Kamemura and Chihara, 2019*; *Murphy and*

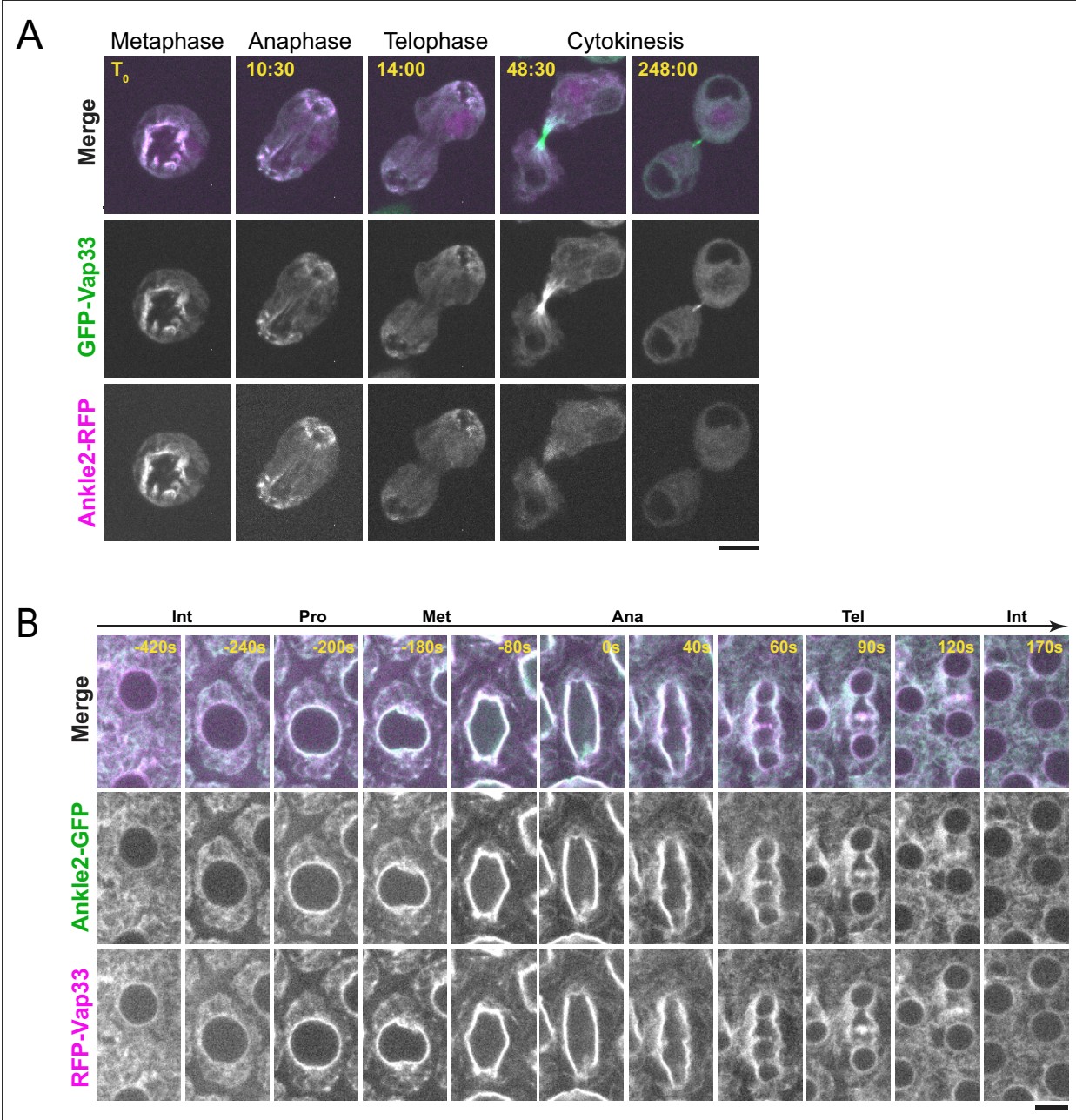

**Figure 3.** Ankle2 co-localizes with Vap33. (**A**) Video images of a D-Mel cell co-expressing Ankle2-RFP and GFP-Vap33 through different stages of cell division. (**B**) Video images of a syncytial embryo co-expressing Ankle2-GFP and RFP-Vap33 through different stages of the cell cycle. Scale bars: 5 μm.

The online version of this article includes the following figure supplement(s) for figure 3:

**Figure supplement 1.** The localization of Ankle2 depends on Vap33 in egg chambers.

*Levine, 2016*). The MSP domain interacts with proteins that contain FFAT motifs (2 phenylalanines in an acidic tract) (*Loewen et al., 2003*). To test if the Vap33 interacts with Ankle2 in this manner, we introduced mutations in the MSP of Vap33 domain predicted to disrupt its ability to bind FFAT motifs based on results with human VAP (K89D, M91D in Vap33) (*Kaiser et al., 2005*). We found that this mutant form of Vap33-Myc was unable to interact with GST-Ankle2$^{910-1174}$ (*Figure 4C*). We then searched for FFAT motifs in Ankle2 using published criteria (*Slee and Levine, 2019*) and identified three candidate motifs (*Figure 4D*). One of them was a particularly good fit. Mutation of this FFAT motif (Fm) abolished the interaction of the Ankle2 C-terminal fragment with Vap33 (*Figure 4B, C*). To test if the interaction occurs in the same way between proteins expressed in *Drosophila* cells, we used

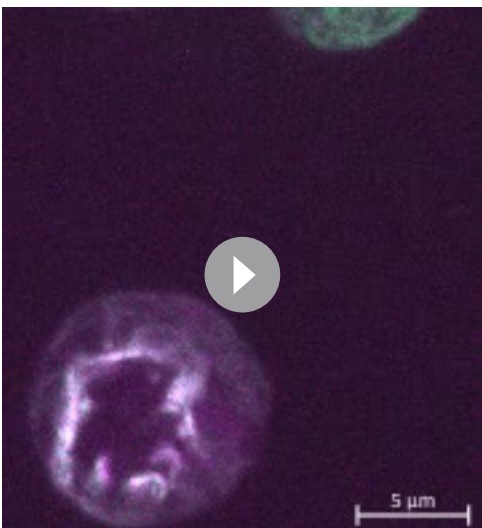

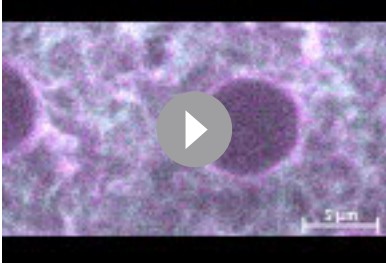

**Video 1.** Localization of Ankle2-RFP and GFP-Vap33 during mitosis and cytokinesis in a D-Mel cell. Orthogonal maximum intensity projection of 4 z-sections spaced by 1 µm. Images were taken every 205 s. Scale bar: 5 µm.
https://elifesciences.org/articles/104233/figures#video1

**Video 2.** Localization of Ankle2-GFP and RFP-Vap33 during the cell cycle in a syncytial embryo. A single plane is shown. Images were taken every 10 s. Scale bar: 5 µm.
https://elifesciences.org/articles/104233/figures#video2

a co-purification assay. We transfected different forms of Ankle2-GFP along with Vap33-Myc and proceeded to GFP affinity purifications followed by western blot for Myc. In this assay, mutation of the first FFAT motif in Ankle2 diminished but did not abolish its interaction with Vap33 (*Figure 4E*). We, therefore, tested the contributions of two other additional FFAT-like motifs (FL1 and FL2, *Figure 4D*). Mutation of FL1 (FL1m) also diminished but did not abolish the interaction of Ankle2 with Vap33, while mutation of FL2 (FL2m) did not perturb the Ankle2-Vap33 interaction. Finally, we found that the combined mutation of the FFAT and FL1 motifs in Ankle2 (Fm +FL1 m) abolished its interaction with Vap33. Thus, we conclude that Vap33, via its MSP domain, can interact with Ankle2 through two redundant FFAT motifs. Interestingly, subsequent modeling using AlphaFold3 predicted that Vap33 interacts with Ankle2 preferentially through a contact between the MSP domain of Vap33 and the FL1 motif of Ankle2 (*Figure 4F*).

## Vap33 is required for the localization of Ankle2 at the ER and reassembling NE

*Drosophila* Vap33 is an essential protein presumably responsible for interactions of the ER with multiple cytoplasmic proteins and structures (*Pennetta et al., 2002*). Nevertheless, we attempted to knock down Vap33 by RNAi in the female germline to examine the effects on Ankle2-GFP localization in embryos. Unsurprisingly, we found that those females were sterile and, therefore, we could not examine embryos. However, we examined ovaries. In control flies, we observed that Ankle2-GFP was weakly localized to nuclear envelopes and strongly localized to the plasma membrane and to ring canals in nurse cells (*Figure 3—figure supplement 1*). By contrast, in Vap33 RNAi ovaries, we found that Ankle2-GFP was largely delocalized from these structures. Whether distinct pools of Ankle2 serve specific functions is a question that remains open. Nevertheless, these results suggested that the localization of Ankle2 depends on Vap33.

To test if Ankle2 requires its interaction with Vap33 for its localization, we generated transgenic flies for the expression of Ankle2^Fm+FL1m^-GFP. We then examined its localization in embryos co-expressing RFP-Vap33. In interphase, Ankle2^Fm+FL1m^-GFP failed to localize to the ER and NE like Ankle2-GFP, instead appearing to diffuse in the cytoplasm (*Figure 4G*, to be compared with *Figure 3B*, and *Video 3*). Surprisingly, as nuclei entered mitosis, Ankle2^Fm+FL1m^-GFP became strongly localized to the nuclear/spindle envelope, similarly to Ankle2-GFP. However, while Ankle2-GFP was retained at those membranous structures in telophase, Ankle2^Fm+FL1m^-GFP disappeared from them, returning to a diffuse, cytoplasmic localization. We conclude that the interaction of Ankle2 with Vap33 is required for the localization of Ankle2 to the ER in interphase, and for its retention at membranes during NE reassembly in telophase.

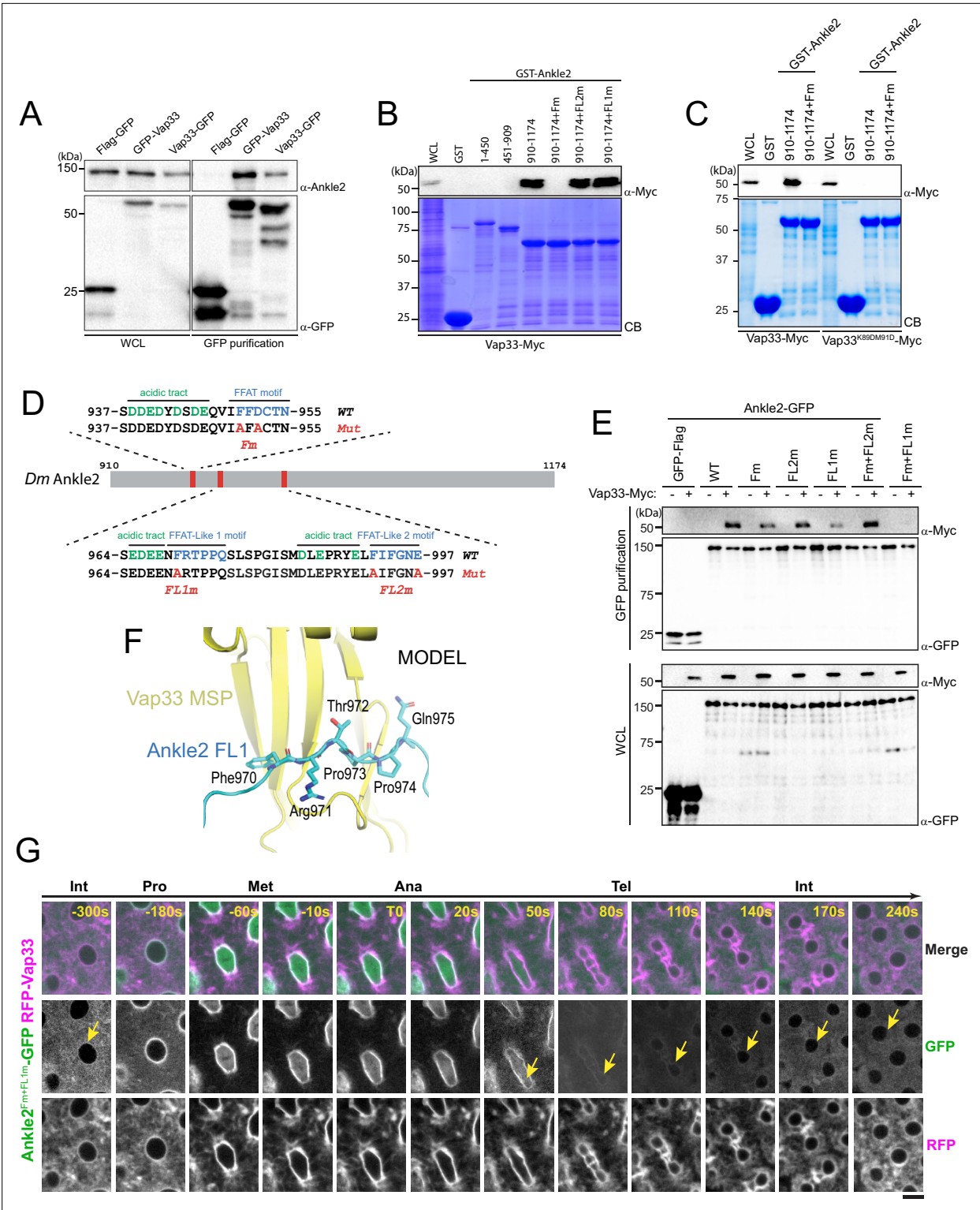

**Figure 4.** Ankle2 interacts with Vap33 through FFAT motifs. (**A**) Cells were transfected with the indicated proteins and used in GFP affinity purifications. Products were analyzed by western blots. WCL: Whole cell extracts. (**B, C**) GST pulldown mapping Ankle2-Vap33 interaction. GST-fused fragments of Ankle2 (wild-type, WT or with the indicated mutations) produced in bacteria were used in GST-pulldowns with extracts from D-Mel cells expressing Vap33-Myc or Vap33^DD-Myc. Pulldown products were analyzed by western blots. CB: Coomassie Blue. WCL: Whole cell extracts. (**D**) FFAT and FFAT-Like motifs identified (green) in the C-terminal end of Ankle2. Acidic residues are in blue. The mutations introduced (Fm, FL1m, and FL2m) are in red. (**E**) The FFAT and FFAT-Like 1 (FL1) motifs contribute to the interaction of Ankle2 with Vap33. Cells were transfected with the indicated proteins and used in

*Figure 4 continued on next page*

*Figure 4 continued*

GFP affinity purifications. Products were analyzed by western blots. (**F**) AlphaFold3 predicted a model of a complex between *Drosophila* Ankle2, Vap33, Mts, and PP2A-29B. The image shows a zoomed-in view of the contact between the MSP domain of Vap33 (yellow) and the FL1 motif of Ankle2 (blue). The full heterotetramer structure model is shown in *Figure 4—figure supplement 1B*. (**G**) The FFAT and FL1 motifs are required for co-localization with Vap33 during telophase and interphase. Video images of a syncytial embryo co-expressing Ankle2$^{Fm+FL1m}$-GFP and RFP-Vap33 through different stages of the cell cycle. To be compared with *Figure 3B*. Note the delocalization of Ankle2$^{Fm+FL1m}$-GFP at the nuclear envelope indicated by the yellow arrows. Scale bars: 5 μm.

The online version of this article includes the following source data and figure supplement(s) for figure 4:

**Source data 1.** Figures with uncropped western blots annotated.

**Source data 2.** Original files for western blots.

**Figure supplement 1.** A Vap33-Ankle2-PP2A complex may promote nuclear reassembly in a localized manner.

**Figure supplement 1—source data 1.** Figures with uncropped western blots annotated.

**Figure supplement 1—source data 2.** Original files for western blots.

Since the ER is the source of the reassembling NE after mitosis, we hypothesized that the interaction of Ankle2 with Vap33 could mediate a localized activity of Ankle2-bound PP2A during nuclear reassembly. In this way, PP2A could dephosphorylate BAF (and potentially other substrates) locally to promote its recruitment near the reforming NE (*Figure 4—figure supplement 1A*). For this model to be valid, Ankle2 should be able to interact with Vap33 and PP2A simultaneously. We used AlphaFold3 to generate structural models of a complex between Vap33, Ankle2, PP2A-29B, and Mts (*Figure 4—figure supplement 1B*). The most probable models show that Ankle2 can simultaneously interact with Vap33, PP2A-29B and Mts without steric clash. Our attempts to detect a complex containing Vap33 and PP2A by simple co-IP and Westerns in D-Mel cells gave variable and inconclusive results. However, we detected Vap33 in complex with PP2A-29B-GFP purified from embryos (*Figure 1E*), despite the fact that PP2A-29B-GFP did not appear clearly enriched at the nuclear/spindle envelope in embryos (*Figure 4—figure supplement 1C*). To test if we could promote the formation of the complex, we overexpressed Ankle2-RFP in cells expressing PP2A-29B-GFP and Vap33-Myc. We found that more Vap33-Myc was co-purified with PP2A-29B-GFP under these conditions, compared with cells without Ankle2-RFP overexpression (*Figure 4—figure supplement 1D*). This co-purification reflected the specific interaction of Ankle2 with Vap33-Myc because it was abrogated with Vap33$^{DD}$-Myc. Consistent with this result, we found that PP2A-29B-GFP became visibly enriched at the NE upon overexpression of Ankle2-RFP in D-Mel cells (*Figure 4—figure supplement 1E*). These results suggest that a Vap33-Ankle2-PP2A complex can mediate the recruitment of a pool of PP2A at the NE.

## Requirements of Ankle2 interactions and motifs for nuclear reassembly

To test the requirements of Ankle2 for its functions in mitosis, we used a rescue assay in D-Mel cells in culture. We depleted endogenous Ankle2 by RNAi and expressed different forms of RNAi-insensitive Ankle2-GFP under the copper-inducible metallothionein promotor (*Figure 5A*). Depletion of Ankle2 alone resulted in nuclear defects including dispersed or aggregated Lamin, fragmented nuclei, DNA devoid of Lamin, and hypercondensed DNA (*Figure 5B, C*). We previously documented these cellular phenotypes, which we showed to occur as a result of defective BAF recruitment to chromosomes and BAF-dependent nuclear reassembly after mitosis (*Li et al., 2024*). In addition, western blotting using Phos-tag revealed that BAF became hyperphosphorylated upon Ankle2 depletion (*Li et al., 2024*). We verified that the expression of Ankle2-GFP rescued the nuclear defects and BAF hyperphosphorylation (*Figure 5B–D*). When examining cellular phenotypes, we scored only GFP-positive cells. The incomplete restoration of unphosphorylated BAF levels upon expression of Ankle2-GFP (WT and

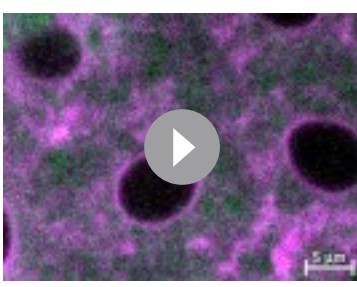

**Video 3.** Localization of Ankle2$^{Fm+FL1m}$-GFP and RFP-Vap33 during the cell cycle in a syncytial embryo. A single plane is shown. Images were taken every 10 s. Scale bar: 5 μm.

https://elifesciences.org/articles/104233/figures#video3

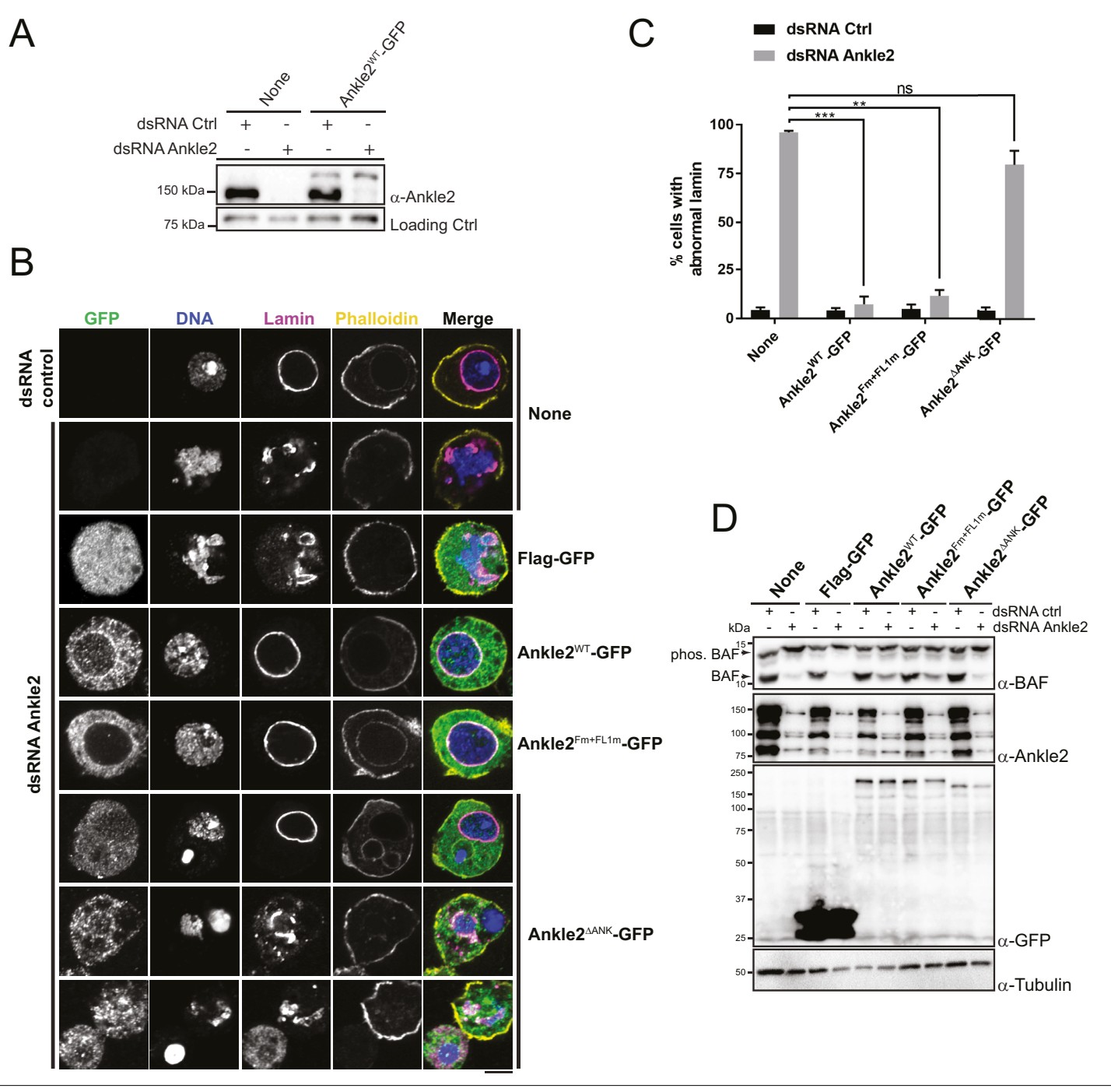

**Figure 5.** Ankle2 interaction with Protein Phosphatases 2A (PP2A) but not with Vap33 is required for nuclear reassembly. (**A**) RNAi depletion of endogenous Ankle2 and simultaneous expression of RNAi-insensitive Ankle2-GFP. D-Mel cells expressing RNAi-insensitive Ankle2-GFP or not were transfected with dsRNA against Ankle2 or non-target dsRNA (NT, against bacterial KAN gene). Four days later, cells were analyzed by western blots. Non-specific band is used as the loading control. (**B**) Immunofluorescence on cells expressing the indicated proteins (right labels) and transfected with the indicated dsRNAs (left labels). GFP fluorescence, DAPI staining for DNA, Lamin (Lamin B) immunostaining, and Phalloidin staining for actin are shown. Red frames indicate the presence of nuclear reassembly defects. Scale bars: 5 µm. (**C**) Quantification of the abnormal Lamin phenotypes from experiments as in B. Note that Ankle2^WT-GFP and Ankle2^Fm+FL1m-GFP, but not Ankle2^ΔANK-GFP, rescue nuclear defects. Averages of three experiments are shown. All error bars: S.D. **p<0.01, ***p<0.001, ns: non-significant from paired t-tests. (**D**) Western blot analysis of experiment shown in B, C. The blot at the top was obtained from a gel containing Phos-Tag to increase the resolution between phosphorylated BAF (phos. BAF) and unphosphorylated BAF (BAF). Note that BAF is hyperphosphorylated after Ankle2 depletion and that this is rescued by the expression of Ankle2^WT-GFP and Ankle2^Fm+FL1m-GFP, but not by Ankle2^ΔANK-GFP. Numerical data are available in *Figure 5—source data 1*.

*Figure 5 continued on next page*

*Figure 5 continued*

The online version of this article includes the following source data and figure supplement(s) for figure 5:

**Source data 1.** Numerical data is used to make graphs.

**Source data 2.** Figures with uncropped western blots annotated.

**Source data 3.** Original files for western blots.

**Figure supplement 1.** Results of rescue experiments in D-Mel cells in culture.

**Figure supplement 1—source data 1.** Numerical data is used to make graphs.

**Figure supplement 1—source data 2.** Figures with uncropped western blots annotated.

**Figure supplement 1—source data 3.** Original files for western blots.

**Figure supplement 2.** Summary of results for various deletion mutants of Ankle2-GFP assessed for their ability to rescue abnormal Lamin phenotypes.

**Figure supplement 3.** Localization of Ankle2-GFP variants during the cell cycle in D-Mel cells.

Fm +FL1 m) in Ankle2-depleted cells is likely due to the inefficient expression of the GFP-tagged protein in a fraction of the cells.

To begin to map the essential regions of Ankle2, we tested the ability of Ankle2$^{1-587}$-GFP (N-terminal half) or Ankle2$^{588-1174}$-GFP (C-terminal half) to rescue the nuclear phenotypes. We found that both truncated proteins were unable to rescue nuclear defects and BAF hyperphosphorylation, suggesting that each half of Ankle2 contains at least one essential region for its function (*Figure 5—figure supplement 1*). The N-terminal half of Ankle2 contains the Ankyrin domain region necessary and sufficient for Ankle2 interaction with PP2A (*Figure 1*). We found that Ankle2$^{\Delta ANK}$-GFP was unable to rescue nuclear defects and BAF hyperphosphorylation (*Figure 5B–D*). These results indicate that the Ankyrin domain is required for the essential function of Ankle2 in BAF dephosphorylation and nuclear reassembly, and suggest that the ability of Ankle2 to interact with PP2A is necessary for this function.

To map more precisely the C-terminal region(s) of Ankle2 required for its function, we made smaller truncations and deletions in Ankle2-GFP, guided in part by sequence conservation between species. We found that deletion of a.a. residues 607–680, 681–937, 938–997, 992–1040, or 1041–1174 partially or completely abrogated the ability of Ankle2-GFP to rescue nuclear defects and BAF hyperphosphorylation (*Figure 5—figure supplements 1 and 2*). The 938–997 segment of Ankle2 contains the FFAT motifs required for the interaction with Vap33 (*Figure 4*). However, we found that the expression of Ankle2$^{Fm+FL1m}$-GFP rescued nuclear defects and BAF hyperphosphorylation (*Figure 5B–D*). These results suggest that the interaction of Ankle2 with Vap33, while it is required for Ankle2 localization to the reassembling NE in embryos, is dispensable for Ankle2 function in BAF dephosphorylation and nuclear reassembly in D-Mel cells in culture.

Disruption of the interaction with Vap33 prevented the localization of Ankle2 at the ER and the reassembling of NE in telophase (*Figure 4G*); however, it did not abolish the localization of Ankle2 at the nuclear/spindle envelope at the early stages of mitosis. We looked for another motif that could contribute to the membrane localization of Ankle2. Human Ankle2 contains a predicted transmembrane (TM) domain in its N-terminus (*Figure 1H*; *Asencio et al., 2012*). Although online prediction tools failed to predict a TM domain in *Drosophila* Ankle2, we noticed a highly hydrophobic segment in its N-terminus (a.a. 14–31). To test if it contributes to the membrane localization of Ankle2, we combined a deletion of the N-terminal 50 a.a. residues (*Figure 1H*) with the FFAT mutations. The resulting Ankle2$^{51-1174\&Fm+FL1m}$-GFP was recruited to the NE during prophase, similarly to Ankle2-GFP and Ankle2$^{Fm+FL1m}$-GFP (*Figure 5—figure supplement 3*). Ankle2$^{51-1174\&Fm+FL1m}$-GFP also largely rescued nuclear defects and BAF hyperphosphorylation, although less efficiently than Ankle2-GFP (*Figure 5—figure supplement 1*). Among various truncated and deleted forms of Ankle2-GFP, only those that lacked C-terminal a.a. residues 938–1174 (Ankle2-GFP 1–587, 1–692 and 1–937) failed to be recruited to the NE during prophase (*Figure 5—figure supplement 3*). This region comprises the 3 candidate FFAT motifs identified. These results suggest that Ankle2 relies on its C-terminus for its NE recruitment during prophase, although the FFAT motifs are not strictly required for this recruitment. We do not know how Ankle2 localizes to the nuclear/spindle envelope independently from Vap33 during the early stages of mitosis, or the importance of this localization.

Overall, we conclude that in addition to its N-terminal PP2A-interacting Ankyrin domain, Ankle2 requires the integrity of its C-terminal portion for its essential function in nuclear reassembly. The

C-terminus of Ankle2 mediates its interaction with Vap33, which is crucial for the localization of Ankle2 to the reassembling of NE in telophase. However, our results suggest that the Ankle2/Vap33 interaction is not strictly required for Ankle2's function in D-Mel cells.

## Interactions of Ankle2 with PP2A and Vap33 promote BAF recruitment to chromosomes during nuclear reassembly

To test the requirements of Ankle2 interactions in vivo, we expressed Ankle2-GFP transgenes in the female germline and early embryos and simultaneously knocked down endogenous Ankle2 using the maternal matα4-GAL-VP16 driver (*Figure 6—figure supplement 1*). RFP-BAF was co-expressed. We then imaged mitosis and monitored the dynamics of the fluorescent proteins (*Figure 6A* top, *Video 4*). As previously observed, RFP-BAF was enriched at the nuclear/spindle envelope during interphase and early mitosis, likely due to its interactions with LEM-Domain transmembrane proteins (*Emond-Fraser et al., 2023*). RFP-BAF also largely colocalized with Ankle2-GFP. After anaphase, RFP-BAF transferred to chromatin, becoming visible at the inner core region of the reassembling nuclei, where the chromatin periphery intersects spindle microtubules (*Figure 6B*), consistent with the role of BAF in nuclear reassembly.

To test the importance of the Ankle2-PP2A interaction, we imaged embryos expressing Ankle2$^{\Delta ANK}$-GFP. We found that RFP-BAF recruitment to the inner core of reassembling nuclei was strongly diminished and delayed compared to embryos expressing Ankle2$^{WT}$-GFP (*Figure 6A* middle, *Video 5* and *Figure 6C*). Given that BAF recruitment promotes NE reassembly and Ankle2 is a NE-associated protein, we examined whether the recruitment of Ankle2$^{\Delta ANK}$-GFP to the inner core region of the reassembling NE was compromised. Fluorescence quantification of NE-associated proteins at the inner core region allows monitoring of NE reassembly, as this region of the reassembling NE is clearly distinct from the spindle envelope in embryos (*Emond-Fraser et al., 2023*; *Figure 6B*, *Figure 6—figure supplement 2*). We found that the recruitment of Ankle2$^{\Delta ANK}$-GFP to the core region was delayed (*Figure 6D*), suggesting that the delay in BAF recruitment caused a delay in NE reassembly. Interestingly, RFP-BAF and Ankle2$^{\Delta ANK}$-GFP were both retained longer at the spindle envelope in embryos expressing Ankle2$^{\Delta ANK}$-GFP, compared with embryos expressing Ankle2$^{WT}$-GFP (*Figure 6E, F*). We conclude that the Ankyrin domain, required for the ability of Ankle2 to interact with PP2A, is necessary for the timely recruitment of BAF at reassembling nuclei and ensuing NE reassembly.

To test the importance of the Ankle2-Vap33 interaction, we imaged embryos expressing Ankle2$^{Fm+FL1m}$-GFP (also depleted of endogenous Ankle2) (*Figure 6A* bottom and *Video 6*). As before, we observed that Ankle2$^{Fm+FL1m}$-GFP failed to localize to membranes during interphase and telophase, and that it was partially recruited to the nuclear/spindle envelope from prophase to anaphase (*Figure 6—figure supplement 2B*). This result rules out the possibility that Ankle2$^{Fm+FL1m}$-GFP localizes to membranes during early mitosis because of a potential complex with endogenous Ankle2. Quantification confirmed that the interaction of Ankle2 with Vap33 is required for the maintenance of Ankle2-GFP at the spindle envelope during telophase and its recruitment to the inner core region of the reassembling NE, at the time when BAF is normally recruited (*Figure 6D, F*). Moreover, we found that the recruitment of RFP-BAF at reassembling nuclei was delayed in embryos expressing Ankle2$^{Fm+FL1m}$-GFP (*Figure 6C*). These results suggest that the ability of Ankle2 to interact with Vap33 contributes to the efficient recruitment of BAF at reassembling nuclei.

## The interactions of Ankle2 with PP2A and Vap33 promote its function in vivo

We examined the importance of Ankle2 and its interactions with embryonic development.

Knockdown of endogenous Ankle2 during late oogenesis using the matα4-GAL-VP16 driver did not significantly diminish the ability of females to lay eggs (*Figure 7A*). However, none of their embryos hatched, indicating a failure in embryonic development in the absence of maternally contributed Ankle2 (*Figure 7—figure supplement 1A, B*). Immunofluorescence revealed that the majority of embryos aborted development in the first mitotic cycle, with a single nucleus/spindle (*Figure 7—figure supplement 1C*). FISH for the X chromosome consistently revealed three foci in the polar body and one or two foci in the mitotic spindle, confirming that the spindle observed corresponded to mitosis and not to meiosis or sperm prior to fertilization (*Figure 7—figure supplement 1D, E*). Expression of RNAi-insensitive Ankle2-GFP in this background fully rescued embryo hatching. This

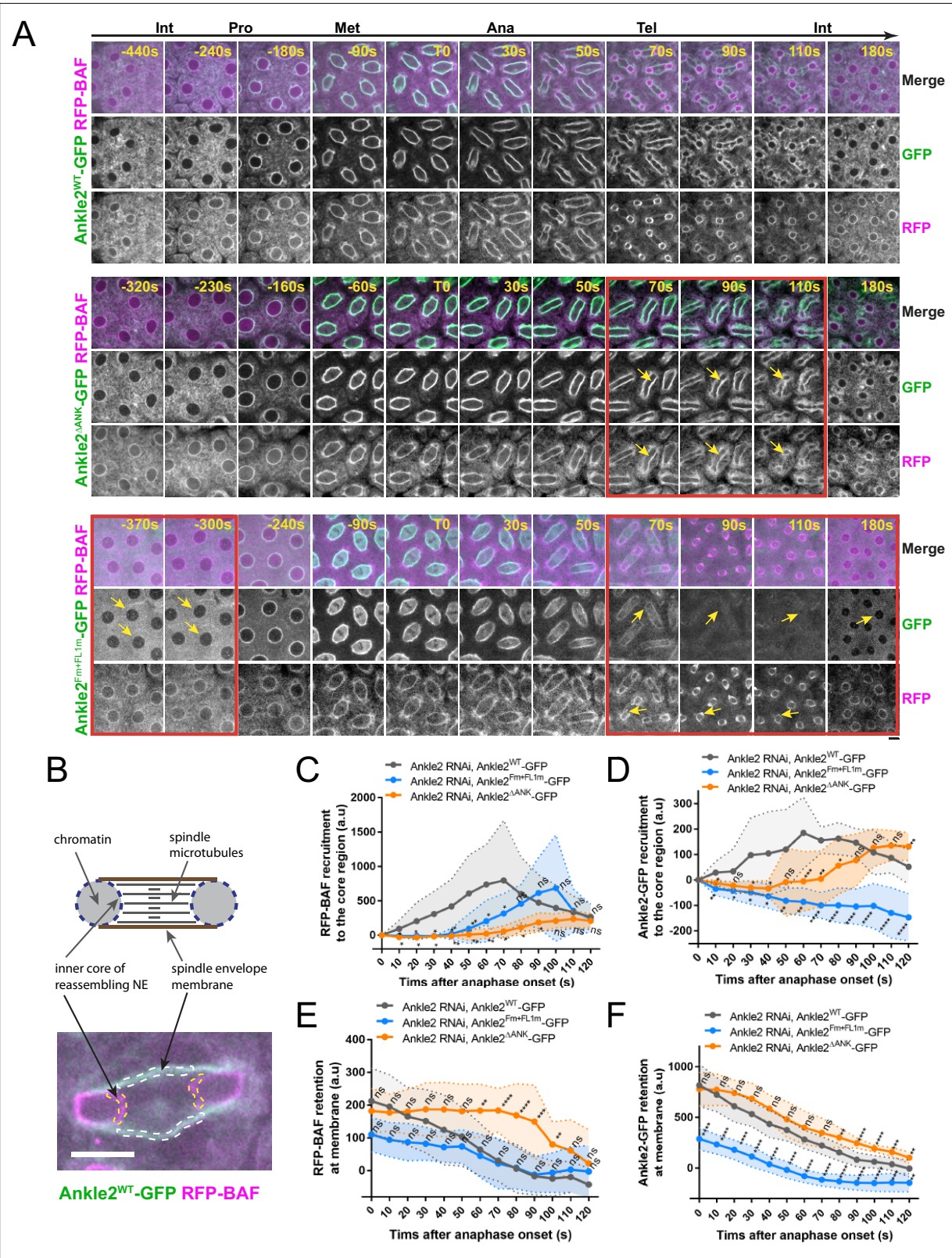

**Figure 6.** Ankle2 interactions with Protein Phosphatases 2A (PP2A) and Vap33 promote BAF recruitment at reassembling nuclei. (**A**) Syncytial embryos depleted of endogenous Ankle2 and expressing RNAi-insensitive Ankle2$^{WT}$-GFP (top), Ankle2$^{\Delta ANK}$-GFP (middle), or Ankle2$^{Fm+FL1m}$-GFP (bottom) along with RFP-BAF were imaged through the cell cycle. Times frames where differences are the most pronounced are highlighted by red frames, along with yellow arrows. (**B**) Illustration of fluorescence quantifications at specific structures. GFP and RFP fluorescence intensities were quantified at the inner

*Figure 6 continued on next page*

*Figure 6 continued*

core region of the reassembling nuclei and at the spindle envelope membranes after anaphase. (**C, D**) Quantification of the recruitment of RFP-BAF (**C**) or Ankle2-GFP variants (**D**) at the inner core region of the reassembling nuclei as a function of time after anaphase onset. (**E, F**) Quantification of the retention of RFP-BAF (**E**) or Ankle2-GFP variants (**F**) at the lateral spindle envelope membranes as a function of time after anaphase onset. Averages of fluorescence intensity from 10 nuclear divisions taken from 5 or 6 embryos are shown per condition. All error bars: S.D. *p<0.05, **p<0.01, ***p<0.001 **** p<0.0001, ns: non-significant from unpaired t-tests. Scale bars: 5 μm. Numerical data are available in *Figure 6—source data 1*.

The online version of this article includes the following source data and figure supplement(s) for figure 6:

**Source data 1.** Numerical data is used to make graphs.

**Figure supplement 1.** Female germline expression of Ankle2-GFP variants in transgenic lines.

**Figure supplement 1—source data 1.** Figures with uncropped western blots annotated.

**Figure supplement 1—source data 2.** Original files for western blots and gels.

**Figure supplement 2.** Quantifications in embryos.

**Figure supplement 2—source data 1.** Numerical data is used to make graphs.

rescue was strongly abrogated when Ankle2$^{\Delta ANK}$-GFP was expressed. The weak rescue observed may be in part attributed to the dilution of Gal4 between two UASp elements (UASp-Ankle2 RNAi and UASp-transgene) because a weak rescue was also observed when GFP alone was expressed (*Figure 7B*). In any case, these results suggest that the interaction of Ankle2 with PP2A, mediated by the Ankyrin domain of Ankle2, is essential during embryo development. Moreover, expression of Ankle2$^{\Delta ANK}$-GFP in the presence of endogenous Ankle2 did not cause a decrease in embryo hatching, ruling out the possibility of toxicity (*Figure 7A*). By contrast, the expression of Ankle2$^{Fm+FL1m}$-GFP fully rescued embryo hatching, suggesting that the interaction of Ankle2 with Vap33 is dispensable for embryo development (*Figure 7B*).

To examine the requirements of Ankle2 interactions in a different context, we turned to wing development. Induction of Ankle2 RNAi in the wing disc pouch using Nubbin-Gal4 (Nub-Gal4) resulted in wingless adults. Simultaneous expression of Ankle2-GFP or Ankle2$^{Fm+FL1m}$-GFP rescued the development of adults with wings of the normal size. However, the expression of Ankle2$^{\Delta ANK}$-GFP resulted in no rescue (*Figure 7C*). These results suggest that the interaction of Ankle2 with PP2A but not with Vap33 is essential for its function during cell proliferation in imaginal wing discs. To test if Vap33 promotes Ankle2 function in a sensitized background, we depleted Ankle2 in the wing pouch using a weaker RNAi line. As a result, smaller adult wings developed (*Li et al., 2024*). We found that the introduction of single mutant alleles of *Vap33* in this background enhances the small wing

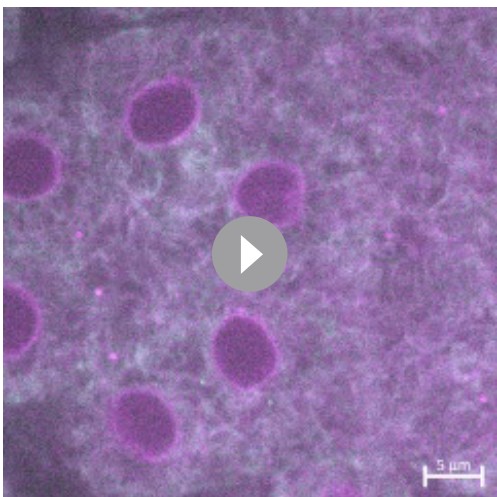

**Video 4.** Localization of Ankle2$^{WT}$-GFP and RFP-BAF during the cell cycle in a syncytial embryo where endogenous is depleted by RNAi. A single plane is shown. Images were taken every 10 s. Scale bar: 5 μm.
https://elifesciences.org/articles/104233/figures#video4

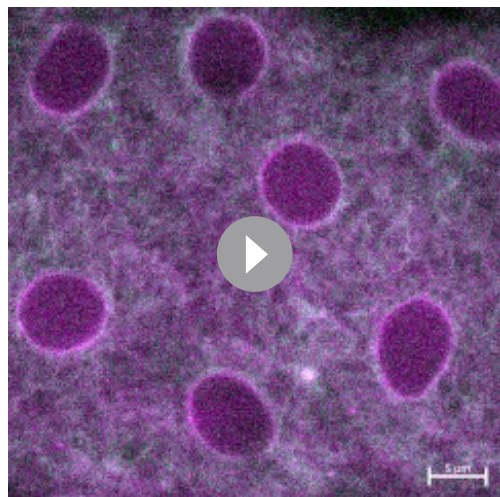

**Video 5.** Localization of Ankle2$^{\Delta ANK}$-GFP and RFP-BAF during the cell cycle in a syncytial embryo where endogenous is depleted by RNAi. A single plane is shown. Images were taken every 10 s. Scale bar: 5 μm.
https://elifesciences.org/articles/104233/figures#video5

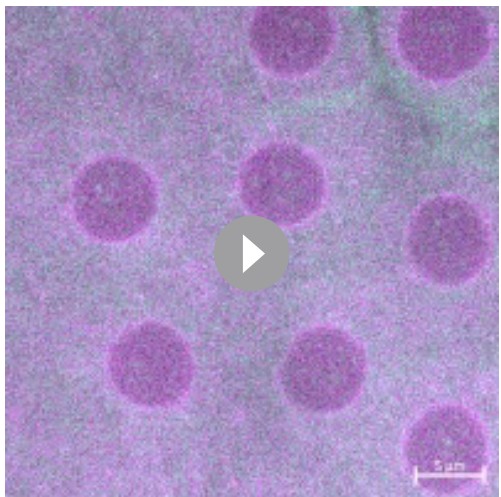

**Video 6.** Localization of Ankle2^Fm+FL1m-GFP and RFP-BAF during the cell cycle in a syncytial embryo where endogenous is depleted by RNAi. A single plane is shown. Images were taken every 10 s. Scale bar: 5 μm.
https://elifesciences.org/articles/104233/figures#video6

phenotype. This result suggests that although the Ankle2-Vap33 interaction is not essential, it nevertheless promotes Ankle2 function during wing development (*Figure 7D*).

## Discussion

Post-mitotic nuclear reassembly is an essential cellular process, yet its molecular underpinnings remain only partially understood. BAF has emerged as a central player required for connections between chromosomes in telophase, for the adhesion of membranes to chromatin through LEM-Domain proteins, and for lamina reassembly (*Haraguchi et al., 2008*; *Haraguchi et al., 2001*; *Li et al., 2024*; *Samwer et al., 2017*). The recruitment of BAF hinges on its dephosphorylation by PP2A, a process dependent on Ankle2 in various animal species including humans, flies, and roundworms (*Asencio et al., 2012*; *Li et al., 2024*; *Mehsen et al., 2018*; *Snyers et al., 2018*). Our study leveraged *Drosophila* as a model to elucidate how Ankle2 functions in this process.

Here, we provide several lines of evidence suggesting that Ankle2 functions as a regulatory subunit of PP2A. We found that Ankle2 uses its Ankyrin domain to interact with PP2A. Ankyrin repeats are known to mediate protein interactions (*Li et al., 2006*). AlphaFold3 predicts an interaction of this domain with PP2A that would position Ankle2 as a regulatory subunit. Indeed, the presence of Ankle2 in the PP2A complex would be mutually exclusive with the presence of known regulatory subunits including B55 (Tws in *Drosophila*) and B56, a hypothesis further supported by our results showing that Tws competes with Ankle2 for binding to PP2A. In addition, Ankle2 co-purified PP2A-29B and Mts without any known PP2A regulatory subunits. Previous work indicated that human Ankle2 uses a similar region to interact with PP2A, comprising part of the Ankyrin domain and a segment immediately preceding it, termed the Caulimovirus Domain (CD, *Figure 1H*; *Asencio et al., 2012*; *Fishburn et al., 2024*). Although the latter region is poorly conserved in *Drosophila*, our modeling suggests that it may participate in the formation of the PP2A-Ankle2 complex (*Fishburn et al., 2024*; *Figure 2C*). Experimental structural determination of the PP2A-Ankle2 is required to reveal the detailed structure of the complex.

Our unbiased phosphoproteomic analysis confirmed that BAF dephosphorylation depends on Ankle2 in *Drosophila* as in *C. elegans* (*Asencio et al., 2012*; *Li et al., 2024*). While this dependence has not been directly tested in human cells, BAF recruitment to segregated chromosomes after anaphase, which is known to require BAF dephosphorylation, requires Ankle2 in HeLa cells (*Asencio et al., 2012*; *Sears and Roux, 2020*; *Snyers et al., 2018*). A LEM domain in human Ankle2 mediates an interaction with BAF (*Snyers et al., 2018*). By contrast, we did not detect an interaction between *Drosophila* Ankle2 and BAF, consistent with the absence of a LEM domain in the former, as for *C. elegans* (*Asencio et al., 2012*; *Fishburn et al., 2024*). Moreover, while human Ankle2 was shown to bind and inhibit the BAF counteracting kinase VRK1 in vitro (*Asencio et al., 2012*), we detected no interaction between Ankle2 and NHK-1/Ballchen (VRK1 ortholog) in *Drosophila*. While a putative interaction between Ankle2 and NHK-1 in *Drosophila* could occur transiently, thereby escaping detection, the simplest interpretation of our results is that the loss of Ankle2 causes BAF hyperphosphorylation by preventing its PP2A-dependent dephosphorylation rather than by preventing inhibition of NHK-1. Our previous genetic analysis in the developing wing suggests that BAF is the most critical substrate of PP2A-Ankle2 during cell proliferation. Indeed, wing developmental defects upon Ankle2 depletion are rescued by lowering the dose of NHK-1 or by expressing an unphosphorylatable mutant form of BAF (*Li et al., 2024*). Ankle2 may enable PP2A to dephosphorylate BAF through a transient interaction independently from the LEM domain and/or by targeting PP2A to a subcellular location that is favorable to the reaction (discussed below). Interestingly, we identified several additional potential

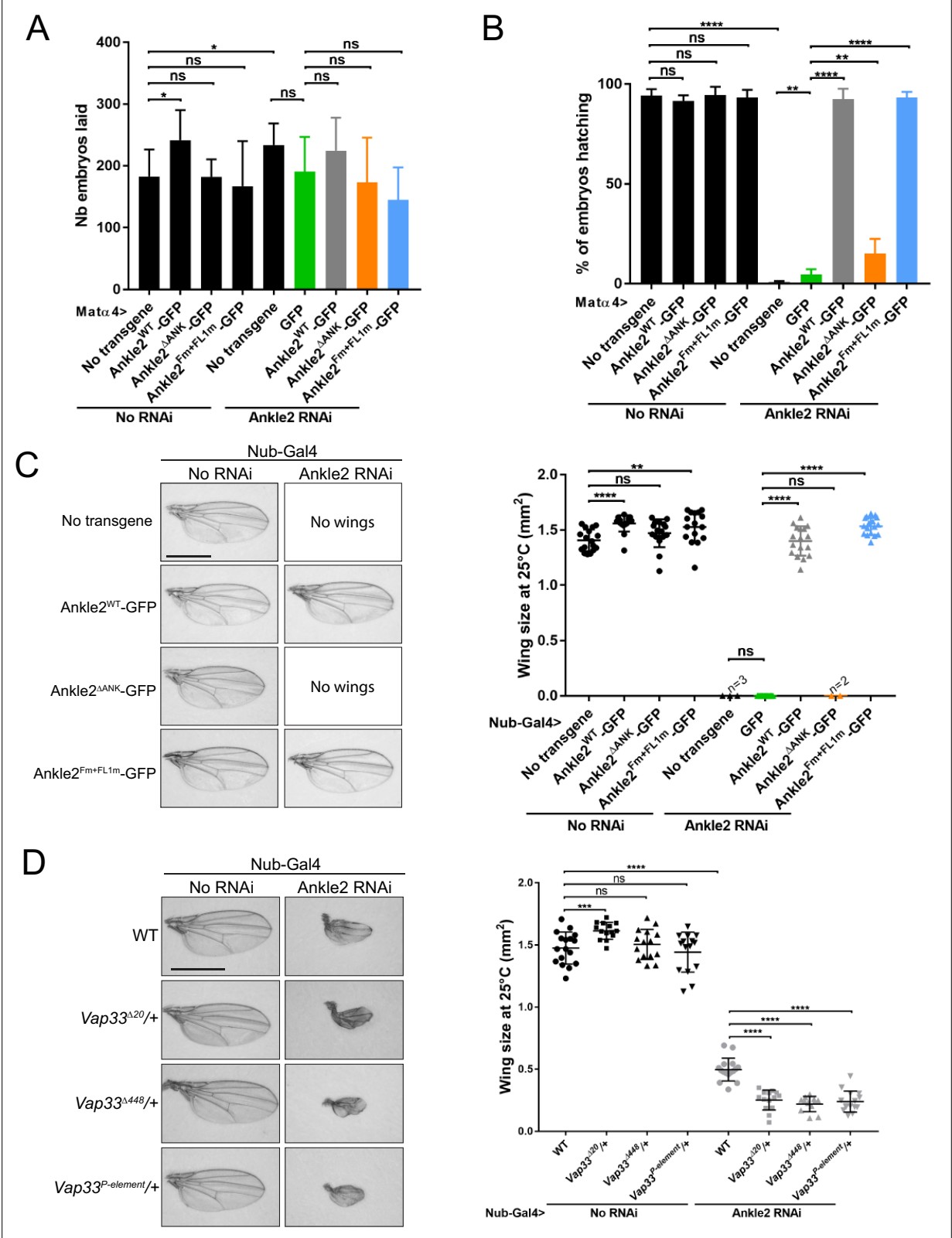

**Figure 7.** Ankle2 interaction with Protein Phosphatases 2A (PP2A) but not with Vap33 is required for development. (**A**) Number of embryos laid over three days per female of the indicated genotypes. (**B**) Percentage of embryos hatching per female of the indicated genotypes. In A-B, the Ankle2 RNAi insertion from line BDSC 77437 was used to deplete endogenous Ankle2 during oogenesis using the maternal matα4-GAL-VP16 driver and RNAi-resistant Ankle2-GFP variants were simultaneously expressed. (**C**) Wing development phenotypes from flies of the indicated genotypes. The Ankle2

*Figure 7 continued on next page*

*Figure 7 continued*

RNAi insertion from line BDSC 77437 was used to deplete endogenous Ankle2 in the wing pouch using Nub-Gal4 and RNAi-resistant Ankle2-GFP variants were simultaneously expressed at 25 °C. Left: Examples of wings. Right: Quantification of wing size. (**D**) Mutations in Vap33 enhance a small-wing phenotype caused by Ankle2 depletion. The weaker Ankle2 RNAi insertion from line VDRC100665 was used to deplete endogenous Ankle2 in the wing pouch using Nub-Gal4. Left: Examples of wings. Right: Quantification of wing size at 25 °C. Averages of 12–17 wings are shown. All error bars: S.D. *p<0.05, **p<0.01, ***<0.001 ****p<0.0001, ns: non-significant from paired t-tests with Welch's correction. Scale bars: 1 mm. Numerical data are available in *Figure 7—source data 1*.

The online version of this article includes the following source data and figure supplement(s) for figure 7:

**Source data 1.** Numerical data is used to make graphs.

**Figure supplement 1.** Phenotypes resulting from RNAi depletion of Ankle2 during oogenesis.

**Figure supplement 1—source data 1.** Numerical data is used to make graphs.

**Figure supplement 1—source data 2.** Figure with uncropped western blot annotated.

**Figure supplement 1—source data 3.** Original file for western blot.

substrates of PP2A-Ankle2 (*Figure 2A*, *Figure 2—source data 1*). We were so far unable to obtain sufficient amounts of the PP2A-Ankle2 complex to conduct robust in vitro phosphatase assays with phosphorylated BAF or other candidate substrates.

We also discovered a novel interaction between Ankle2 and the ER protein Vap33. We show that FFAT motifs in Ankle2 interact with the MSP domain of Vap33. This interaction is required for the localization of Ankle2 to the reassembling NE in telophase, and to the NE and ER in interphase. The recent identification of putative FFAT motifs in human Ankle2 (*Neefjes and Cabukusta, 2021*) suggests that the interaction between Ankle2 and VAP family proteins of the ER may be conserved.

These findings led us to propose a model where Vap33 targets the PP2A-Ankle2 holoenzyme to ER membranes, promoting BAF dephosphorylation and recruitment in a localized manner during nuclear reassembly. Supporting this model, we detected a complex comprising Vap33, Ankle2, PP2A-29B, and Mts. Our results using cells in culture, embryos and developing wings strongly suggest that the Ankle2-PP2A interaction is essential for BAF dephosphorylation and recruitment to nascent nuclei in telophase, as well as for nuclear reassembly and animal development. However, it remains formally possible that the deletion of Ankyrin repeats used to disrupt the Ankle2-PP2A interaction abrogated another, unknown aspect of Ankle2 function. Similarly, we found that the Ankle2-Vap33 interaction, while being less critical, also promotes BAF recruitment during nuclear reassembly in embryos. Conceptually, the novel mechanism we propose implies that the ER is more than a passive source of membranes in nuclear reassembly; it implicates the ER as a carrier of localized enzymatic activity needed for the process.

Future work should examine the functions of other protein interactions and potential substrates of PP2A-Ankle2 that we identified, which could implicate this holoenzyme in the regulation of various cellular processes. Particularly intriguing are the interactions we identified between Ankle2 and Cyclin-CDK subunits. The possibility that Ankle2, while functioning as a PP2A subunit, may also impact Cyclin-CDK functions warrants further investigation. Conversely, Cyclin-CDKs may also contribute to the regulation of PP2A-Ankle2 in the cell cycle, an area that remains to be explored.

## Materials and methods
### Plasmids and mutagenesis

*Drosophila* cells expression vectors were generated using the Gateway recombination system (Invitrogen). The cDNA of interest was first cloned into the pDONR221 entry vector and then recombined into the destination vector with a C-terminal or N-terminal tag containing copper-inducible or constitutively active promoters, pMT, or pAc5, respectively. The RNAi-resistant (RNAi res) cDNA for Ankle2 was generated by replacing codons of the original cDNA of the longest form of Ankle2 with synonymous codons. The following expression vectors were generated: pAc5-FLAG-GFP, pAc5-FLAG-Tws, pMT-Ankle2-GFP, pMT-Ankle2$^{Fm}$-GFP, pMT-Ankle2$^{FL1m}$-GFP pMT-Ankle2$^{FL2m}$-GFP, pMT-Ankle2$^{Fm+FL1m}$-GFP, pMT-Ankle2$^{Fm+FL2m}$-GFP, pMT-Ankle2$^{\Delta ANK}$-GFP, pMT-Ankle2-GFP (RNAi res), pMT-Ankle2$^{\Delta ANK}$-GFP (RNAi res), pMT-Ankle2$^{Fm+FL1m}$-GFP (RNAi res), pMT-Ankle2$^{Fm1+FL2m+\Delta TM}$-GFP (RNAi res), pMT-Ankle2$^{1-587}$-GFP (RNAi res), pMT-Ankle2$^{588-1174}$-GFP (RNAi res), pMT-Ankle2$^{\Delta 992-1040}$-GFP (RNAi

res), pAc5-Ankle2-FLAG (RNAi res), pAc5-Ankle2$^{F1m+FL2m}$-FLAG (RNAi res), pAc5-Ankle2$^{\Delta ANK}$-FLAG(RNAi res), pAc5-Vap33-Myc, pAc5-Vap33$^{87D89D}$-Myc, pAc5-Vap33-GFP, pAc5-Vap33$^{87D89D}$-GFP, pAc5-PP2A-29B-GFP, pAc5-GFP-PP2A-29B, pMT-RFP-BAF$^{WT}$, pMT-RFP-BAF$^{3A}$. Point mutations and deletions in the pDONR of interest were generated using QuickChange Lightning Site-Directed Mutagenesis Kit (Agilent), as described by the manufacturer's instructions. GST-Ankle2 (WT, 1–274, 275–450, 451–909, 910–1174, 910–1174+Fm, 910–1174+FL1 m, 910–1174+FL2 m) expression vectors were constructed into the pGEX4T vector by classic cloning process. pUAS-Ankle2-GFP, pUAS-Ankle2-GFP (RNAi res), pUAS-Ankle2$^{Fm+FL1m}$-GFP (RNAi res), pUAS-Ankle2$^{\Delta ANK}$-GFP (RNAi res), pUAS-PP2A-29B-GFP, pUAS-RFP-BAF, pUAS-RFP-Vap33 and were generated by cloning PCR amplicons of interest into the pUAS-K10attB vector using restriction enzymes, NotI and BamHI.

## Cell lines

D-Mel (D.mel-2) cells from *Drosophila melanogaster* were obtained from the lab of David Glover at the University of Cambridge where they were purchased from Invitrogen (#10831–014) and where VA was a postdoc. Their identity was confirmed by their ability to grow in their dedicated serum-free insect cell media, their expected morphology, and numerous proteomic analyses. These cells tested negative for mycoplasma contamination.

## Cell culture and transfections

Cells were cultured in Express Five medium (Invitrogen) supplemented with glutamine, penicillin, and streptomycin (Wisent) at 25 °C. Transfections with plasmids were performed using X-tremeGENE HP DNA Transfection Reagent (Roche) following the manufacturer's protocol. Stable cell lines were selected by adding 20 ug/ml blasticidin into the cell culture after transient transfections of plasmids of interest. While inducible pMT-based vectors contain the gene coding for blasticidin resistance, pAc5-based vectors were co-transfected with pCoBlast to confer resistance to blasticidin to the cells. 300 µM CuSO$_4$ were added into the medium to induce expression of the plasmids containing pMT promoter, at least 16 hr before use. For RNA interference, dsRNAs were generated from PCR amplicons using a Ribomax Kit (Promega). RNAi non-target (NT) was generated against the sequence of the procaryotic kanamycin resistance gene. 1×10$^6$ cells were plated in a six-well plate and treated with 1 ml of medium containing 20 µg of indicated dsRNA for 4 d (rescue experiments). Cells were then harvested and analyzed by immunoblotting, immunofluorescence, or live-cell imaging.

## Fly genetics

Fly husbandry was done according to standard procedures. Oregon R was used as the WT strain. Transgenic flies for expression of *pUAS-Ankle2-GFP, pUAS-RFP-Vap33,* and *pUAS-PP2A-29B-GFP* were generated by site-directed insertions of the pUAS-K10attB-based vectors on the second chromosome in the attP40 strain (BestGene Inc, Chino Hills, CA, UAS). Transgenic lines for expression of *pUAS-Ankle2-GFP RNAi res* (WT, Fm +FL1 m, ΔANK) were generated by site-directed insertions of the pUAS-K10attB-based vectors on the third chromosome in the attP154 strain (BestGene Inc, Chino Hills, CA, UAS). UAS-Ankle2 RNAi (BDRC 77437) and UAS-GFP (BDRC 4776) used in this study were purchased from Bloomington *Drosophila* Stock Center.

For the genetic rescue experiment, a combination of UAS-Ankle2-GFP (RNAi res, WT, and mutant) and Ankle2 RNAi (BDRC 77437) were generated using a pair of second and third balancer chromosomes. UAS-GFP (BDSC 4776) was used as the control for ruling out the possible effect due to Gal4 dilution. Expression of transgenes in the embryo and developing wings was driven by matα4-Gal4-VP16 (BDRC 7062) and Nubbin-Gal4 (BDRC 86108). All crosses were performed at 25 °C with 60–70% humidity. Fertility tests were carried out by scoring eggs on the grape juice agar from a single female being crossed with three males for 24 hr.

## Protein purifications from *Drosophila* cells and fly embryos

GFP affinity purifications were performed from *Drosophila* cells and fly embryos, as described (*Emond-Fraser et al., 2023*). Briefly, cells stably expressing pAc5-FLAG-GFP, pMT-GFP-Ankle2, and pMT-Ankle2-GFP were harvested from four confluent 175cm2 flasks and resuspended in lysis buffer (75 mM K-HEPES pH7.5, 150 mM NaCl, 2 mM EGTA, 2 mM MgCl$_2$, 1 mM DTT, 10 µg/ml aprotinin, 10 µg/ml leupeptin, 1 mM PMSF, 5% glycerol, 0.5% Triton X-100). Embryos were collected every 2 hr

and dechorionated in 50% bleach, washed in PBS, and frozen in liquid nitrogen. Embryos were then crushed in lysis buffer as above. Cell and embryo lysates were incubated for 20 min at 4 °C on a wheel, centrifuged at max speed for 15 min at 4 °C, and subsequently incubated with GFP-trap nanobeads (Chromotek) for 2 hr. Beads were firstly washed five times with lysis buffer and then washed additionally five times in PBS with protease inhibitors (1 mM PMSF, 10 µg/ml aprotinin, and 10 µg/ml leupeptin) before sending for mass spectrometry. A small portion of the sample was also eluted in Laemmli buffer and analyzed by SDS-PAGE with silver nitrate staining.

## Mass spectrometry

For proteomic analyses, samples were reconstituted in 50 mM ammonium bicarbonate urea 1 M with 10 mM TCEP [Tris(2-carboxyethyl)phosphine hydrochloride; Thermo Fisher Scientific], and vortexed for 1 hr at 37 °C. Chloroacetamide (Sigma-Aldrich) was added for alkylation to a final concentration of 55 mM. Samples were vortexed for another hour at 37 °C. One microgram of trypsin was added, and digestion was performed for 8 hr at 37 °C. Samples were dried down and solubilized in 5% acetonitrile (ACN)–4% formic acid (FA). The samples were loaded on a 1.5 µl pre-column (Optimize Technologies, Oregon City, OR). Peptides were separated on a home-made reversed-phase column (150 µm i.d. by 200 mm) with a 56 min gradient from 10 to 30% ACN-0.2% FA and a 600 nL/min flow rate on an Easy nLC-1200 connected to an Exploris 480 (Thermo Fisher Scientific, San Jose, CA). Each full MS spectrum acquired at a resolution of 120,000 was followed by tandem-MS (MS-MS) spectra acquisition on the most abundant multiply charged precursor ions for 3 s. Tandem-MS experiments were performed using higher energy collision dissociation (HCD) at a collision energy of 34%. The data were processed using PEAKS X Pro (Bioinformatics Solutions, Waterloo, ON) and a Uniprot database. Mass tolerances on precursor and fragment ions were 10 ppm and 0.01 Da, respectively. Fixed modification was carbamidomethyl (C). Variable selected posttranslational modifications were acetylation (N-ter), oxidation (M), deamidation (NQ), and phosphorylation (STY). The data were visualized with Scaffold 5.0 (protein threshold, 99%, with at least 2 peptides identified and a false-discovery rate [FDR] of 1% for peptides).

For phosphoproteomic analyses, 500 µg of cell lysate (measured by Bradford assay) were reconstituted in 50 mM ammonium bicarbonate with 10 mM TCEP [Tris(2-carboxyethyl)phosphine hydrochloride; Thermo Fisher Scientific], and vortexed for 1 hr at 37 °C. Chloroacetamide (Sigma-Aldrich) was added for alkylation to a final concentration of 55 mM. Samples were vortexed for another hour at 37 °C. 10 µg of trypsin was added, and digestion was performed for 8 hr at 37 °C. Samples were dried down in a speed-vac. For the $TiO_2$ enrichment procedure, sample loading, washing, and elution were performed by spinning the microcolumn at 8000 rpm at 4 °C in a regular Eppendorf microcentrifuge. The spinning time and speed were adjusted as a function of the elution rate. Phosphoproteome enrichment was performed with $TiO_2$ columns from GL Sciences. Digests were dissolved in 400 µL of 250 mM lactic acid (3% TFA/70% ACN) and centrifuged for 5 min at 13,000 rpm, and the soluble supernatant was loaded on the TiO2 microcolumn previously equilibrated with 100 µL of 3% TFA/70% ACN. Each microcolumn was washed with 100 µL of lactic acid solution followed by 200 µL of 3% TFA/70% ACN to remove nonspecific binding peptides. Phosphopeptides were eluted with 200 µL of 1% NH4OH pH 10 in water and acidified with 7 µL of TFA. Eluates from $TiO_2$ microcolumns were desalted using Oasis HLB cartridges by spinning at 1200 rpm at 4 °C. After conditioning with 1 mL of 100% ACN/0.1% TFA and washing with 0.1% TFA in water, the sample was loaded, washed with 0.1% TFA in water, then eluted with 1 mL of 70% ACN (0.1% TFA) prior to evaporation on a SpeedVac. The extracted peptide samples were dried down and solubilized in 5% ACN-0.2% FA. The samples were loaded on an Optimize Technologies $C_4$ precolumn (0.3 mm i.d. by 5 mm) connected directly to the switching valve. They were separated on a home-made reversed-phase column (150 µm i.d. by 150 mm Phenomenex Jupiter C18 stationary phase) with a 120 min gradient from 10 to 30% ACN-0.2% FA and a 600 nL/min flow rate on an Easy nLC-1200 (Thermo Fisher Scientific, San Jose, CA) connected to an Exploris 480 (Thermo Fisher Scientific, San Jose, CA). Each full MS spectrum was acquired at a resolution of 120,000 and followed by tandem-MS (MS-MS) spectra acquisition for 3 s on the most abundant multiply charged precursor ions. Tandem-MS experiments were performed using higher-energy collisional dissociation (HCD) at a collision energy of 27%. The data were processed using PEAKS X Pro (Bioinformatics Solutions, Waterloo, ON) and a *Drosophila melanogaster* unreviewed Uniprot database. Mass tolerances on precursor and fragment ions were 10 ppm and 0.01 Da, respectively. Variable selected posttranslational modifications were carbamidomethyl (C), oxidation

(M), deamidation (NQ), acetyl (N-term), and phosphorylation (STY). The data were visualized with Scaffold 5.0 (protein threshold, 99%, with at least two peptides identified and a false-discovery rate [FDR] of 1% for peptides).

The mass spectrometry proteomics data have been deposited to the ProteomeXchange Consortium via the PRIDE (*Perez-Riverol et al., 2022*) partner repository with the dataset identifier PXD059738 (https://doi.org/10.6019/PXD059738).

## Western blotting and immunofluorescence

Protein lysates were analyzed on SDS-PAGE and then transferred onto a PVDF membrane. The membrane was then blocked with 5% of milk solution for 1 hr to prevent non-specific binding. Subsequently, the membrane was probed with primary antibodies for 3 hr at 25 °C or overnight at 4 °C. The following primary antibodies used in this study were: anti-GFP from rabbit (1:5000, Torrey Pine Biolabs), anti-Myc 9E10 from mouse (1:2000, #sc-40, Santa Cruz Biotechnology), anti-FLAG M2 from mouse (1:2000, #F1804, Sigma), anti-tubulin DM1A from mouse (1:5000, #T6199, Sigma), anti-Ankle2 from rabbit (1:1000, custom-made by Thermo Fisher Scientific), anti-Mts from mouse (1:1000, #610555, BD Biosciences), anti-Tws from rabbit (1:2000, custom-made by Thermo Fisher Scientific). After 3 washes of 10 min with PBS containing 0,1% Tween (PBST, 0,1%), the membrane was probed with a secondary antibody for 30 min at room temperature. Peroxidase-conjugated anti-rabbit or anti-mouse from goat (1:5000, Jackson ImmunoResearch) were used as secondary antibodies. The membrane being washed 3 times in PBST, 0,1% was then incubated with Clarity Western ECL Substrate (# 170–5061, Bio-Rad) and subsequently imaged using ChemiDoc MP Imaging system.

To evaluate the phosphorylation level of the protein of interest, protein lysates were separated by SDS-PAGE in the presence of the Phos-tag (Fujifilm WAKO Chemical), following the manufacturer's instructions. One wash of the acrylamide gel with the transfer buffer containing 1 mM EDTA and subsequent second wash with only the transfer buffer were performed before the electrotransfer.

For Immunofluorescence, cells were first fixed with PBS containing 4% formaldehyde for 25 min. After three washes of 10 min with PBS containing 0,2% Triton X-100(PBST), cells were permeabilized and blocked in PBST containing 1% BSA for 1 hr. Then, cells were incubated with primary antibodies for 2 hr at room temperature, washed three times with PBST, and subsequently incubated with secondary antibodies and DAPI for 1 hr avoiding light. Primary antibodies used in this study were: anti-Lamin from mouse (1:200, DSHB Hybridoma Product ADL84.12), and anti-GFP from rabbits (1:1000, Torrey Pine Biolabs). Secondary antibodies from mouse and rabbit were respectively coupled to Alexa-647(1:200, Invitrogen) and Alexa-488 (1:200, Invitrogen).

## Fluorescence in situ hybridization (FISH)

*Drosophila* embryos were collected, dechorionated in 50% bleach, and washed three times in 0,7% NaCl, and 0,05% Triton-100. Eggs were then fixed in methanol: heptane (1:1) and rehydrated successively in methanol: PBS solutions of the ratio of 9:1, 7:3, and 1:1. FISH was performed following procedures as described in, using a probe against the 359-base pair peri-centromeric repeat on the X chromosome. α-Tubulin in the embryos was stained with primary antibodies (YL1/2 from rat at 1:2000, Sigma) and subsequent secondary antibodies (anti-rat Alexa 647 at 1:1000, Invitrogen). DNA was marked with QUANT-IT Oli-green at 1:5000 (#O7582, Invitrogen). Tetrahydronaphthalene was used to mount embryos before being imaged by a confocal microscope.

## GST pulldowns

D-Mel cells expressing Ac5-Vap33-Myc, Ac5-Vap33[87D89D]-Myc, Ac5-PP2A-29B-GFP or Ac5-GFP-PP2A-29B were harvested from a confluent 175 cm² flask. Cells were then lysed in lysis buffer (75 mM K-HEPES pH 7.5, 150 mM KCl, 2 mM EGTA, 2 mM MgCl₂, 5% glycerol, 0.2% Triton X-100, 1 mM DTT) supplemented with protease inhibitors cocktail as described above at 4 °C for 20 min. Cell lysates being centrifuged at max speed for 15 min were incubated with Sepharose beads bound to GST and GST-Ankle2 (truncated forms) for 1.5 hr at 4 °C. Beads were washed five times in lysis buffer and subsequently used for SDS-PAGE analysis and western blot.

## Co-immunoprecipitation (Co-IP)

Pelleted cells from confluent 25 cm$^2$ flasks were lysed in 20 mM Tris-HCl pH 7.5, 150 mM NaCl, 2 mM MgCl2, 0.5 mM EDTA, 1 mM DTT, 5% glycerol, 0.5% NP40 Substitute supplemented with protease inhibitors cocktail as described above. Cell lysates being lysed for 15 min on a rotating wheel at 4 °C were centrifuged at max speed for 10 min, and supernatants were incubated with 10 µl GFP-Trap agarose beads (Chromotek) for 2 hr at 4 °C. Beads were washed four times in 20 mM Tris-HCl pH 7.5, 150 mM NaCl, 2 mM MgCl2, 0.5 mM EDTA, 1 mM DTT, 5% glycerol, and 0.1% NP40 Substitute with protease inhibitors. Samples were eluted in Lammeli buffer and analyzed using western blot.

## Microscopy

A confocal system Leica SP8 was used for imaging of fixed cells and embryos. Acquired images of cells were subsequently processed with the Lightning system. Live imaging of cells and embryos was performed by a spinning disk confocal system (Yokogawa CSU-X1 5000) mounted on a fluorescence microscope (Zeiss Axio Observer.Z1). Cells were cultured on a LabTek II chambered coverglass (#155409, Thermo Fisher Scientific) at least 2 hr before time-lapse imaging. For live analysis of embryos, 0–2 hr embryos were dechorionated in 50% bleach and then aligned on a coverslip (#P35G-1.5–14 C, MatTek) and covered with halocarbon oil. Films of embryos were taken every 10 s with six confocal sections of about 3 µm. To monitor Ankle2-GFP (WT and mutants) and RFP-BAF levels, respectively, the mean intensity of GFP and RFP fluorescence in the region of interest were measured directly with Zen software (Zeiss) at different time points. For each point, the mean intensity of GFP and RFP was normalized by subtraction of mean intensity at anaphase onset (T0) in the core region. Images of adult flies were acquired by stereomicroscope with a camera (Canon). Developing wings of adult flies were dissected and quantified using Fiji software (National Institute of Health).

## Structure predictions

Prediction of co-complexes was performed using the AlphaFold3 (*Abramson et al., 2024*) AlphaFold Server (Beta) (https://alphafoldserver.com), with default seed auto-generation and with seed provided, yielding sets of 5 models. Co-complexes modeled are of Mts (NCBI accession NP_001285724.1), PP2A-29B (NP_001027225.1) and Ankle2 (NP_573221.2); Mts, PP2A-29B and Tws (NP_001287269.1); and Mts, PP2A-29B, Ankle2 and Vap33 (NP_570087.1). Each set of 5 models showed self-consistent interactions and a typical example is shown in the figures.

## Statistical analysis

All the graphs and statistical analysis in this study were done by GraphPad software. In all figures, the results of quantifications are expressed as mean ± SD with indicated statistics in the legend. Overall, p-values are represented as follows: *$p<0.05$, **$p<0.01$, ***$p<0.001$, ****$p\leq0.0001$, and n.s. (not significant) is $p>0.05$.

## Materials availability statement

All reagents newly generated in this study are available upon request.

# Acknowledgements

This work was funded by a Project Grant from the Canadian Institutes of Health Research (CIHR) to VA (175132). JL received a studentship from the Fonds de Recherche du Québec – Santé (FRQS). TMS is a member of the Centre de recherche en biologie structurale, funded by FRQS Research Centres Grant #288558. We thank Christian Charbonneau for his precious help with the microscopy.

## Additional information

### Funding

| Funder | Grant reference number | Author |
|---|---|---|
| Canadian Institutes of Health Research | 175132 | Vincent Archambault |
| Fonds de Recherche du Québec - Santé | | Jingjing Li |
| Fonds de Recherche du Québec - Santé | 288558 | T Martin Schmeing |

The funders had no role in study design, data collection and interpretation, or the decision to submit the work for publication.

### Author contributions

Jingjing Li, Conceptualization, Resources, Formal analysis, Investigation, Methodology, Writing – original draft, Writing – review and editing; Xinyue Wang, Resources, Formal analysis, Funding acquisition, Investigation, Methodology; Laia Jordana, Mohammed Bourouh, Resources, Formal analysis, Investigation; Éric Bonneil, Formal analysis, Investigation, Writing – original draft; Victoria Ginestet, Resources, Investigation; Momina Ahmed, Cristina Mirela Pascariu, Investigation; T Martin Schmeing, Formal analysis, Investigation, Visualization, Writing – original draft, Writing – review and editing; Pierre Thibault, Resources, Supervision; Vincent Archambault, Conceptualization, Resources, Formal analysis, Supervision, Funding acquisition, Investigation, Methodology, Writing – original draft, Project administration, Writing – review and editing

### Author ORCIDs

Jingjing Li ⓘ https://orcid.org/0009-0006-2060-5260
Vincent Archambault ⓘ https://orcid.org/0000-0002-2857-7667

Reviewer #1 (Public review): https://doi.org/10.7554/eLife.104233.3.sa1
Reviewer #2 (Public review): https://doi.org/10.7554/eLife.104233.3.sa2
Reviewer #3 (Public review): https://doi.org/10.7554/eLife.104233.3.sa3
Author response https://doi.org/10.7554/eLife.104233.3.sa4

## Additional files

### Supplementary files

MDAR checklist

### Data availability

The mass spectrometry proteomics data have been deposited to the ProteomeXchange Consortium via the PRIDE partner repository with the dataset identifier PXD059738 (https://doi.org/10.6019/PXD059738). Source date files have been provided for Figures 1, 2, 5, 6, 7 and Figure supplements of Figures 1, 5, 6, 7.

The following dataset was generated:

| Author(s) | Year | Dataset title | Dataset URL | Database and Identifier |
|---|---|---|---|---|
| Li J, Wang X, Jordana L, Bonneil E, Ginestet V, Ahmed M, Bourouh M, Pascariu CM, Schmeing TM, Thibault P, Archambault V | 2025 | Mechanisms of PP2A-Ankle2 dependent nuclear reassembly after mitosis | http://www.ebi.ac.uk/pride/archive/projects/PXD059738 | PRIDE, PXD059738 |

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
